



# Annual wake impacts in and between wind farm clusters modelled by a mesoscale numerical weather prediction model and fast-running engineering models

Sara Porchetta[1,2], Michael F. Howland[1], Ruben Borgers[3], Sophia Buckingham[4], and Wim Munters[5]

[1]Massachusetts Institute of Technology, Civil and Environmental Engineering, 77 Massachusetts Avenue, 1-290, Cambridge, MA 02139, United States of America
[2]Faculty of Civil Engineering and Geosciences, Delft University of Technology, Delft, The Netherlands
[3]Department of Earth and Environmental Sciences, KU Leuven, Belgium
[4]ENGIE Laborelec, 1630 Linkebeek, Belgium
[5]von Karman Institute for Fluid Dynamics, 1640 Sint-Genesius-Rode, Belgium

**Correspondence:** s.p.porchetta@tudelft.nl

**Abstract.**

With the rapid increase in wind farm developments, it is essential to evaluate the impacts of newly constructed wind farms on adjacent existing and planned wind farms. Various numerical models have been used to study the interactions between adjacent farms, spanning from fast-running engineering models to numerical weather prediction models. These models are

essential to anticipate and mitigate wake effects in future wind energy deployments. Since the atmospheric conditions are variable over the year, it is important to characterize the variation of the wake interactions over the year, important for wind farm operation and planning. Due to the higher computational cost of numerical weather prediction models compared to fast-running engineering models, they are limited in their capacity to evaluate a wide range of design scenarios or very long simulation periods. In this study, we investigate the annual variation of wake effects coming from a new wind farm cluster

on adjacent, existing wind farms in the North Sea using a simulation of a representative year. We compare results from a numerical weather prediction model (the Weather and Research Forcasting model) and for multiple fast-running engineering wake models (Gauss-BPA, Jensen ($k_w$ = 0.02), Jensen ($k_w$ = 0.04), Cumulative curl, TurbOPark ($A$ = 0.04), TurbOPark ($A$ = 0.06)), providing insights into variations in wake loss predictions among the models. Indeed, both the numerical weather prediction model and the engineering models make different assumptions to predict wake interactions between wind farms.

Throughout this study a distinction is made between external wake losses, caused by the newly built wind farm cluster only, and internal wake losses, which are generated by the individual wind farms of the existing cluster. Temporal variations in stability are the main driver of hourly and seasonal variations in external wake losses, while internal losses are also determined by seasonal variations in wind speed. While yearly averaged external wind farm losses from the numerical weather prediction model are limited to 4%, the internal wake losses reach as high as 36% for the closest adjacent, existing wind farm to the

new wind farm cluster. Additionally, all engineering models considered predict lower wake losses compared to the numerical weather prediction model, but predictions exhibit a very large range of magnitudes, ranging from 98% to 33% difference for external wakes and 59% to 14% for internal wakes compared to the numerical weather prediction model. Not only do the results





differ quantitatively but also qualitatively between model strategies, i.e. yearly spatial distributions, especially for the external wake predictions of the former fast-running engineering models (Gauss-BPA, Jensen ($k_w = 0.02$), Jensen ($k_w = 0.04$)) and the numerical weather prediction model. These engineering models do not capture the same qualitative trend as the numerical weather prediction model while the newly designed engineering models (Cumulative curl, TurbOPark ($A = 0.04$), TurbOPark ($A = 0.06$)) do. For the internal wake losses, qualitatively, both engineering models and the numerical weather prediction model show higher internal wake losses for turbines located in the center of the wind farm, with highest losses for densely spaced turbines, and lower losses at the edge of the wind farm, however all models show different magnitudes of losses.

## 1 Introduction

Wind energy holds significant potential as a primary contributor to meet the growing demand for clean energy in the global community, with the expectation that wind power will account for up to one-third of the global energy supply by 2050 (Veers et al., 2019). Following the signing of the Ostend Declaration in April 2023, Belgium, Germany, Denmark, and the Netherlands have committed to deliver 65 GW of offshore wind by 2030, illustrating the exponential growth one can expect in development areas such as the North Sea. As the deployment of wind farms continues to accelerate, leading to lower wind farm spacing in between wind farms, it becomes crucial to investigate interactions between neighboring wind farms. Wind farms create wakes, which are regions of momentum and energy deficit downwind of the turbines (Fischereit et al., 2022b). These wakes can extend over large distances (Platis et al., 2018; Schneemann et al., 2020) and influence the power production of downwind wind farms (Lundquist et al., 2019; Barthelmie and Jensen, 2010; Pryor et al., 2021; Stieren and Stevens, 2022).

A variety of numerical models have been employed to examine these farm-farm wakes, each with a focus on different atmospheric scales, ranging from the microscale to the mesoscale, synoptic scales, and even global scales (Veers et al., 2019; Porté-Agel et al., 2020). Microscale numerical approaches for studying farm-farm wakes encompass a range of fidelity levels. High fidelity computational fluid dynamics models such as large eddy simulations (Maas and Siegfried, 2022; Stieren and Stevens, 2022) and mid fidelity Reynolds-Averaged Navier-Stokes models (van der Laan et al., 2015, 2023) are computationally expensive, limiting their application to specific atmospheric conditions. On the other hand, low fidelity fast-running engineering wake models provide a cost-effective alternative for wake studies which can account for multiple atmospheric conditions (Nygaard et al., 2020; Nygaard and Hansen, 2016; Munters et al., 2022). Recently, the significance of studying wakes within a mesoscale (or numerical weather prediction) model framework has gained prominence (Cuevas-Figueroa et al., 2022). Mesoscale models are particularly valuable for their ability to incorporate the influence of multiple atmospheric variables, which may exhibit non-uniform spatial and temporal distributions. However, it is important to note that typical horizontal resolutions in mesoscale models for wind energy research often range from 1 to 5 km (Fischereit et al., 2022a), which is larger than the size of typical wind turbines. To account for subgrid scale effects of wind turbines which are not explicitly resolved by the model, wind farm parameterizations are employed within mesoscale models (Veers et al., 2019). Over the years, different wind farm parameterizations have been developed and used to investigate a wide range of questions. These investigations have included studying the effects of wind veer and shear (Redfern et al., 2019), wind-waves and swell (Porchetta et al., 2021),



wind farm layout (Pan and Archer, 2018), and model resolution (Pryor et al., 2020), among others on the power production and wakes of wind turbines.

So far, mesoscale wake studies have predominantly focused on limited time periods, that are not sufficient to directly evaluate annually averaged wake impacts or annual wake variability. These shorter time periods have been considered due to the high computational costs. Pryor et al. (2021) investigated eleven 5-day periods aiming to represent seasonality. A similar approach was used by Fischereit et al. (2020) where 180 days were selected to represent the wind and wave climate to investigate wind-wave-wake interactions of wind farms in the North Sea. More recently Borgers et al. (2024) investigated farm-farm wakes for one year for different wind farm density scenarios for the planned wind farm locations in the North Sea. Their findings revealed that farm-farm wakes are highly sensitive to capacity density, inter-farm spacing, and wind farm sizes. Also Akhtar et al. (2021) simulated the wake impacts in the North Sea over multiple years, however, both Akhtar et al. (2021) and Borgers et al. (2024) employed similar wind turbines over the full area and did not account for the actual type of installed/planned wind turbines. These theoretical studies can provide useful insight into the importance of farm-farm interactions but are less relevant to quantify the effect of future planned wind farm clusters on the actual adjacent existing wind farms, as these studies do not consider the actual wind farm characteristics. In contrast, Cuevas-Figueroa et al. (2022) investigated farm-farm interaction of an operational farm for shorter periods of time, and stated that yearly runs are still computational expensive. Lastly, Rosencrans et al. (2024) studied the sub annual variability of offshore wind deployments in the mid-Atlantic and observed monthly and daily fluctuations in power deficits resulting from farm-farm interactions. They also showed the non-negligible effect of farm-farm interactions. It is clear that there is a growing interest on farm-farm interaction in numerical weather prediction models for periods that resolve seasonal variations.

Computationally less costly are fast-running engineering wake models commonly used as a tool to estimate yearly production of wind farms and thus for wake loss estimates (Peña et al., 2018). Not only does the computational cost differ between the different models (numerical weather prediction models and fast-running engineering models) they furthermore differ in their assumptions made and thus the complexity to represent the wakes of the wind farm. While previous comparisons between these two models exist (Fischereit et al., 2022b; Hansen et al., 2015), they have not been conducted over an extended time period. Fischereit et al. (2022b) found that mesoscale models are capable of accurately estimating the mean wind speed deficit but incur higher error with estimating peak wind speed deficits. On the other hand, most engineering wake models perform well for intra-farm wakes but less so for farm-farm wakes, as they indicate a faster recovery of wind-speed deficits. Fischereit et al. (2022b) suggest that engineering wake models could be enhanced in estimating farm-farm wakes with the availability of more calibration data. Recently, new developments have been proposed to reduce the underestimation of fast-running engineering models to better represent farm-farm wakes (Pedersen et al., 2022; Bay et al., 2023).

In this study, we aim to investigate the annual averaged wake effects and the sub-annual wake variability of a newly planned wind farm cluster on adjacent existing wind farms. We predict these effects using a numerical weather prediction model and fast-running engineering models, each offering different fidelity levels and we compare their results. Both models are suitable for longer-term analyses but a yearly comparison is so far missing. The comparison is tested for the new Princess Elisabeth (PE) wind farm cluster which will be built in the near future in the Belgian part of the North Sea. The investigated year corresponds



to the most meteorologically representative year of the past 30 years, allowing us to examine the most typical wake effects on the existing neighboring wind farms, which are important for the operations of the latter. Furthermore, this study focuses solely on commonly used wind farm parameterization and engineering wake models using standard calibration parameters. Despite the knowledge that certain models/calibration parameters perform better under specific conditions, the primary objective of this study is to offer insights into the annual wake impact and sub-annual variability using the most commonly out-of-the-box employed methods.

Examining wake interactions with numerical models is of paramount importance when evaluating the effects of newly constructed wind farms. Given the lower wind farm spacing, encountering wake effects from neighboring wind farms is virtually inevitable in the near future. Therefore, it is crucial to compare the prediction range of existing numerical wake models with different fidelity levels in estimating both sub-annual and in the future potentially inter annual wake variations. This comparison can help us get an estimate of the uncertainty of the prediction of farm-farm wake modelling. Therefore, the study aims to provide a quantitative and qualitative comparison of the planned Prinses Elisabeth wind farm cluster across different fidelities of numerical wake models, stemming from mesoscale weather prediction models to fast-running engineering models.

## 2 Material and methods

The materials and methods presented in this section are applied to the new Princess Elisabeth (PE) wind farm cluster and the existing adjacent wind farms. However, the same methodology can be extrapolated to study other farm-farm interactions.

### 2.1 Study area and period

Within the southern North Sea, a new wind farm cluster, the Princess Elisabeth cluster (PE) (red dots in Fig. 1), will be built. Although the final installed capacity is still under discussion, ranging from 3.15 GW to 3.5 GW, only one possible wind farm scenario (Munters et al., 2022) will be considered in this study, corresponding to a newly installed capacity of 3.5 GW with 15 MW turbines with a rotor diameter of $D$=236 m and a hub height of $z_h$ =146 m. While we investigate the impact of the new wind farm cluster on all neighboring farms, the impact on the Belgian-Dutch (BE-NL) wind farm cluster, located just downstream of the main southwesterly wind direction, is the primary focus in this study. However, as other nearby wind farms are likely to be affected (Lundquist et al., 2019; Barthelmie and Jensen, 2010; Pryor et al., 2021; Stieren and Stevens, 2022), they are also considered in the numerical weather prediction model of this study (Fig. 1). In total, the influence of the Princess Elisabeth wind farm zone on 1409 wind turbines is investigated. However, for the fast-running engineering models only the Belgian-Dutch wind farm cluster is considered. Details of the included wind turbines can be found in Table A1. The turbines are modeled in the numerical simulations with the appropriate turbine manufacturer curves or approximations thereof (Pierrot, 2024; EMD International A/S). The locations of the existing wind turbines are extracted from Hoeser et al. (2022), while the locations for the new Princess Elisabeth wind farm are from Munters et al. (2022) and Cornillie et al. (2021). The latter layout results from an even distribution of turbines over the lease area, accounting for a known exclusion zone, without delving into the specifics of micrositing based on sea surface topography.



The simulations are performed for the year 2016 as it is a representative year for the long-term wind climate at 100 m above mean sea level for the study area. This was determined based on the methodology outlined in Borgers et al. (2024), based on hourly ERA5 data for the period 1990-2020. This methodology identifies a representative year by comparing the wind speed and wind direction distributions for a single year with the long-term distributions. Using a representative year ensures that this study can provide a good basis for long-term representation of the influence of building the new Princess Elisabeth wind farm cluster close to the neighbouring wind farms.

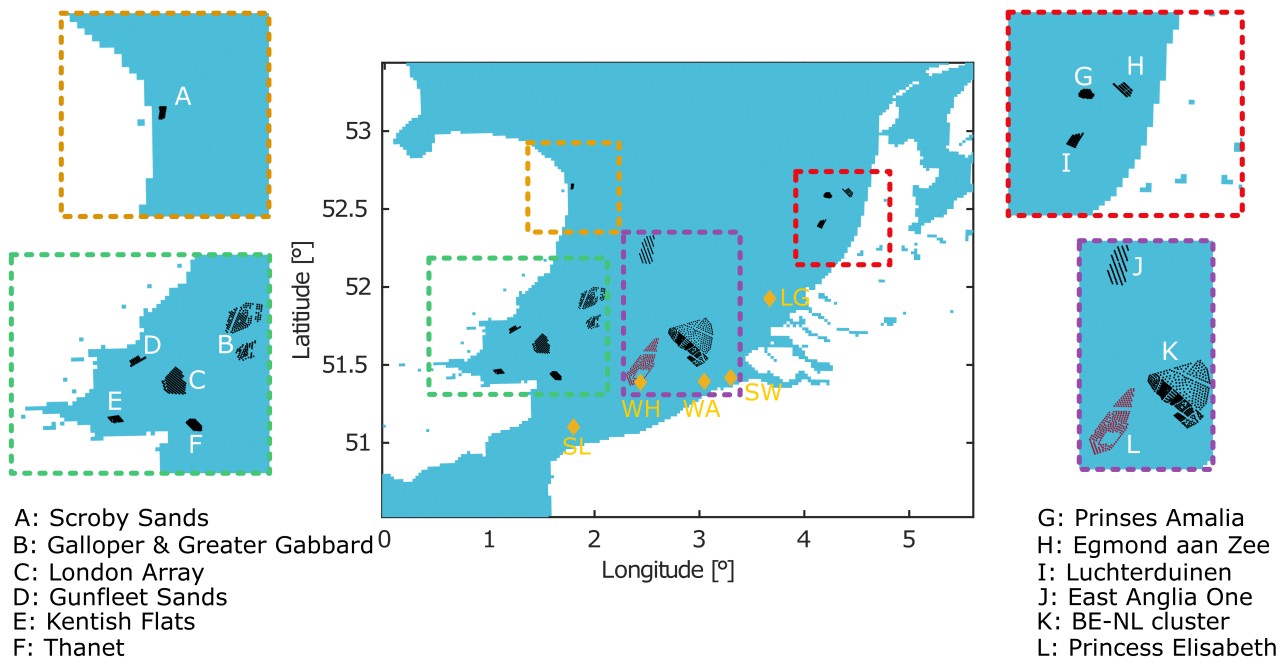

A: Scroby Sands
B: Galloper & Greater Gabbard
C: London Array
D: Gunfleet Sands
E: Kentish Flats
F: Thanet

G: Prinses Amalia
H: Egmond aan Zee
I: Luchterduinen
J: East Anglia One
K: BE-NL cluster
L: Princess Elisabeth

**Figure 1.** Location of the new Princess Elisabeth wind farms (red dots) and the existing wind farms (black dots) within the southern North Sea. Yellow diamonds show the location of measurement masts (Table 1)

## 2.2   Observation stations

The simulation results are compared to observations of offshore measurements. For the year 2016, 5 measurement platforms (yellow diamonds Fig. 1, Table 1) were measuring wind speed and wind directions at altitudes ranging from 14 m to 38 m with a temporal resolution of 10 minutes (Borgers et al., 2024). These measurements were taken by boom- or platform-mounted anemometers and wind vanes. At the location of Wandelaar both measurement techniques are present.



**Table 1.** Observational mast name, location and measurement altitude.

| Observational mast | Longitude | Latitude | Height [m] |
|---|---|---|---|
| Westhinder (WH) | 2.4378° E | 51.3883° N | 26 |
| Wandelaar (WA) | 3.0458° E | 51.3945° N | 26 |
| Scheur-Wielingen (SW) | 3.2985° E | 51.4183° N | 25 |
| Sandettie-Lightship (SL) | 1.800° E | 51.102° N | 14 |
| Lichteiland-Goeree (LG) | 3.67° E | 51.9258° N | 38 |

## 2.3 Mesoscale model setup

135  The numerical model used is the Weather Research and Forecasting (WRF) model v4.3 (Skamarock et al., 2008) with three
nested domains (150 × 150, 190 × 190 and 220 × 190 grid cells) with a horizontal grid spacing of 18 km for the outermost
domain, 6 km for the middle domain and 2 km for the innermost domain (Fig. 2), following the recommendation of Fischereit
et al. (2022a). The vertical resolution has 80 vertical levels which are stretched, resulting in more levels close to the surface,
and follows the recommendations of Lee and Lundquist (2017). Within the first 150 m there are approximately 15 vertical

140  levels. The model top is located at 1000 Pa. The Global Multi-resolution Terrain Elevation Data (GMTED2010) is imposed
for the geographical data with a resolution of 30 arcseconds. The initial and boundary layer conditions come from the ERA5
reanalysis data set provided by the European Centre for Medium-Range Weather Forecast (Hersbach et al., 2018a, b), which
have a 0.25° x 0.25° horizontal grid resolution and 1 h temporal resolution and 37 pressure levels up to 1 hPa. The physical
parameterizations in this study are based on previous studies for similar applications on the same geographical region (Table

145  2) (Siedersleben et al., 2020; Porchetta et al., 2020, 2021). The parameterizations are a double-moment microphysics scheme
(Morrison et al., 2005), the long and short wave Rapid Radiative Transfer Model for General Circulation Models (RRTMG)
(Iacano et al., 2008), the Mellor-Yamada Nakanishi Ninno (MYNN) planetary boundary layer scheme (Nakanishi and Niino,
2006) for which the turbulent advection is turned on for the wind turbines (Archer et al., 2020), the revised MM5 surface layer
scheme (Jiménez et al., 2012), the Noah land surface scheme (Tewari and Cuenca, 2004) and the Kain-Fritsch cumulus scheme

150  (Kain, 2004) (only applied to the parent domain as the two innermost domains have a convective permitting resolution). If
wind turbines are simulated, the wind farm parameterization of Fitch et al. (2012) is used.



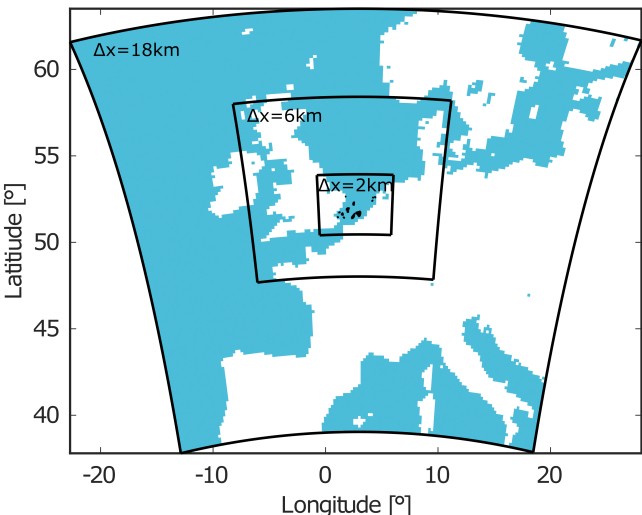

**Figure 2.** The WRF computational configuration with 3 nested domains with a horizontal grid spacing of 18 km, 6 km and 2 km for the outer to the innermost domain. The innermost domain shows all wind farm clusters included in the simulations (see Fig.1 and Table 1 for details).

**Table 2.** Parameterizations used in the WRF simulations.

| Physics | Parameterization | Reference |
|---|---|---|
| Mircophysics | double-momentum | Morrison et al. (2005) |
| Short and long wave radiation | RRTMG | Iacano et al. (2008) |
| Planetary boundary layer | MYNN | Nakanishi and Niino (2006) |
| Surface layer | revised M5 | Jiménez et al. (2012) |
| Land surface | Noah | Tewari and Cuenca (2004) |
| Cumulus | Kain-Fritsch | Kain (2004) |
| Wind farm | Fitch | Fitch et al. (2012) |

Three simulations for the year 2016 are performed, a first one where no wind farms are included, which is used as a reference simulation. One simulation with only the existing wind farms (black dots Fig. 1) and one simulation with the existing wind farms and the new Princess Elisabeth wind farms (black and red dots Fig. 1). The names of these three simulations throughout the paper will be referred to as, 'no WF', 'without PE' and 'with PE' respectively. The simulations including wind turbines (without PE and with PE) have exactly the same set-up as the no WF setup, grid size and employed parameterizations except for the wind farm parameterization.

155



## 2.4 Engineering wake model setup

The FLORIS v3.4 software framework[1] is used to perform the fast-running engineering wake model simulations. FLORIS is an
open-source modular framework that holds a wide range of velocity deficit models, wake combination models, and turbulence
models. It was originally conceived as an efficient wake model for studies oriented to wind-farm control (Gebraad et al., 2016;
Annoni et al., 2016, 2018), but has since emerged as a popular modeling framework also for general wake and yield assessments
(van Beek et al., 2021; Munters et al., 2022; Doekemeijer et al., 2022).

An ensemble of the most used classical and novel wake deficit models is employed in this study to provide a quantitative and
qualitative comparison with the mesoscale model. More specifically, the Jensen model (Katic et al., 1986) and the Gaussian
model proposed by Bastankhah and Porté-Agel (2014) (further referred to as Gauss-BPA) are included as the long-term in-
dustry standard and the legacy FLORIS default option respectively. While the latter is used with the out-of-the-box calibration
parameters, for the Jensen model we include two options for the wake decay coefficient $k_w$, i.e. a 'standard' value of 0.04 and
a reduced value of 0.02 following recent recommendations on a large-scale benchmark over an entire fleet of offshore wind
farms (Nygaard et al., 2022). In addition, we include two more recently proposed models, namely the Cumulative Curl model
(Bay et al., 2023) which was designed for large wind farm arrays, and the TurbOPark model (Pedersen et al., 2022) that was
tailored to mitigating the excessive wake recovery in long-distance wake development (including farm-farm interactions) with
a modified turbulence recovery scheme. While the Cumulative Curl model is run with default hyperparameters, for TurbOPark
we include the default expansion coefficient parameter $A = 0.04$ as well as the value $A = 0.06$ as proposed by van der Laan
et al. (2022), who claim that the default parameter tends to overestimate wake effects. Turbulence intensity input is set at a
typical 6% value for the offshore North sea area, and overlapping wakes are combined in quadrature (except for the Cumulative
Curl model, which has an integrated combination procedure). A summary of the deficit models used in this study is provided
in Table 3.

The fast-running engineering model simulations are performed in line with a typical energy yield assessment. A look-up
table of turbine power and losses is created by discretizing the wind rose into 72 sectors (5 degrees) for wind speeds between 0
and 35 ms$^{-1}$ with a resolution of 1 ms $^{-1}$, and driving each model with the resulting homogeneous wind fields. A time series
is constructed by linking the resulting look-up table to the time series extracted from the no WF WRF run (detailed above).
Effects of blockage are not explicitly modeled. Similar to the WRF case setup, to isolate effects of the PE wind farm cluster
on the existing zone, for every engineering wake model two simulations are performed, i.e., one with and one without the PE.
Further, a no-wake scenario is also computed by combining the driving wind speed from no WF WRF with the turbine power
curve.

---

[1] Available at: https://github.com/NREL/floris/releases/tag/v3.4





**Table 3.** Overview of simulated cases with engineering models

| Model | Hyperparameter | Reference |
|---|---|---|
| Gauss-BPA | default | Bastankhah and Porté-Agel (2014) |
| Jensen | $k_w = 0.04$ | Katic et al. (1986) |
| Jensen | $k_w = 0.02$ | Katic et al. (1986), hyperparam. following Nygaard et al. (2022) |
| Cumulative curl | default | Bay et al. (2023) |
| TurbOPark | $A = 0.04$ (default) | Pedersen et al. (2022) |
| TurbOPark | $A = 0.06$ | Pedersen et al. (2022), hyperparam. following van der Laan et al. (2022) |

## 2.5 External and internal wake loss quantification

In this study, we seek to quantify contributions to total wake losses that are internal and external. External wake losses are wake power losses caused by the new PE wind farm cluster on the existing wind farms. External wake losses are calculated using Eq. (1), based on Eqs. (2) and (3) and are largely stemming from wake interactions from neighboring wind turbines within one wind farm internally. Internal wake losses are the wake power losses caused by the existing wind farm itself. Internal wake losses are calculated based on Eq. (2). Here $P_{\mathrm{withoutPE}}$, $P_{\mathrm{withPE}}$ represent the power produced by the turbines excluding and including the PE wind farm, respectively. $P_{\mathrm{noWF}}$ is the hypothetical power the wind turbines would have produced if there were no wake effects (internal or external). This is calculated for every turbine by taking the wind speed at hub height for the no WF WRF simulation and using the turbine manufacturer curves.

$$\mathrm{Loss_{external}} = \mathrm{Loss_{internal+external}} - \mathrm{Loss_{internal}} \tag{1}$$

$$\mathrm{Loss_{internal}} = 100 - \left( \frac{P_{\mathrm{withoutPE}}}{P_{\mathrm{noWF}}} \right) \times 100 \tag{2}$$

$$\mathrm{Loss_{internal+external}} = 100 - \left( \frac{P_{\mathrm{withPE}}}{P_{\mathrm{noWF}}} \right) \times 100 \tag{3}$$

## 2.6 Stability classification

Because atmospheric stability has significant effects on the wakes of offshore wind turbines (Platis et al., 2022; Rosencrans et al., 2024; Wu et al., 2023), it is important to investigate how the wake impacts of the new PE wind farm are influenced by different atmospheric stability regimes. The atmospheric stability in this work is defined by the Monin-Obukhov length ($L$), which categorizes atmospheric stability based on the ratio of turbulence production by mechanical shear to turbulence production by buoyancy Monin and Obukhov (1954). The different stability classes based on $L$ are shown in Table 4; this





classification is proposed by Van Wijk et al. (1990) and has been utilized in various studies (Porchetta et al., 2019; Nybø et al., 2020; Platis et al., 2022; Howland et al., 2022). The Monin-Obukhov length is output at every 10-minute time interval by WRF and is calculated by the surface layer scheme (Jiménez et al., 2012).

**Table 4.** Atmospheric stability classification based on the Monin-Obukhov length ($L$).

| Stability class | Range of L |
|---|---|
| Very stable | $0 \leq L < 200$ |
| Stable | $200 \leq L < 1000$ |
| Near neutral | $1000 \leq |L|$ |
| Unstable | $-1000 < L \leq -200$ |
| Very unstable | $-200 < L \leq 0$ |

## 3   Results and discussion

First the simulation results of the mesoscale model are discussed (Section 3.1-3.5), after which a comparison with the fast-
running engineering models follows (Section 3.6).

### 3.1   WRF validation

This work does not aim at reaching the optimal representation of the wind field by selecting the best model setup through a sensitivity study on grid size and/or employed parameterizations as already has been investigated by Li et al. (2021); Siedersleben et al. (2020); Hahmann et al. (2014); Vermuri et al. (2022); Tomaszewski and Lundquist (2022). However, it is crucial to vali-
date the fields of interest, even though reaching the highest precision is not the primary focus of this research. For this purpose, wind speed and wind direction from the no WF WRF simulations is compared against observed values at five different locations (Table 1). The bias (simulated minus observed) is presented in Table 5 for both wind speed and wind direction. In 2016 only few (Throntonbank I, II and III, Belwind and Northwind) of the Belgian-Dutch wind farms were operational (182 turbines of the 572 turbines). The inclusion of wind farms in the simulation causes wakes at the observational mast locations, thereby
deviating from the observed values. This is why the verification of the model is only done for the no WF WRF simulation.

    The bias in the simulation without wind farms (no WF) is relatively small, ranging from -0.47 to 1.22 ms$^{-1}$ for wind speed and -3.37° to 12.15° for wind direction. This falls within the range reported in previous studies on offshore wind fields (Rosencrans et al., 2024; Fischereit et al., 2022b; Borgers et al., 2024; Li et al., 2021). Examining specific observation locations, namely Westhinder, Sandettie-Lightship, and Lichteiland-Goeree, the wind speed predicted by WRF tends to underestimate
observed values offshore, consistent with findings by Li et al. (2021), Rosencrans et al. (2024) and Borgers et al. (2024). In contrast, at Wandelaar and Scheur-Wieligen, closer to the coast, the WRF model tends to overestimate wind speed (Hahmann et al., 2014), suggesting a potential coastal effect not well captured by WRF. Due to the relatively coarse coastal resolution, WRF sees a longer fetch which results in higher wind speeds in the model (Hahmann et al., 2014). No clear trend is evident



in biases concerning the altitude of the considered measurement masts, except that the highest observation point exhibits the
lowest bias (Klemmer et al., 2024), which is the opposite of what was found by Sward et al. (2023); Munoz-Esparza et al.
(2012).

**Table 5.** Wind speed and wind direction bias (simulation-observations) for the no WF WRF simulation at the five different observation
locations (Table 1).

| Observational mast | Wind speed bias [ms$^{-1}$] | Wind direction bias [°] |
|:---:|:---:|:---:|
| | no WF | no WF |
| Westhinder (WH) | -0.33 | 3.37 |
| Wandelaar (WA) US | 0.49 | 12.15 |
| Wandelaar (WA) vane | 0.59 | 4.85 |
| Scheur-Wielingen (SW) | 1.22 | 8.53 |
| Sandettie-Lightship (SL) | -0.47 | 4.88 |
| Lichteiland-Goeree (LG) | -0.05 | 3.77 |

## 3.2 Annual energy production and mean yearly wind speed deficit

The annual energy production (AEP) and mean yearly wind speed deficit before (top row Fig. 3) and after constructing the
new Princess Elisabeth wind farm cluster (middle row Fig. 3) are shown in Fig. 3. The wind speed deficit is defined by the
yearly mean wind speed of the simulation with or without PE minus the yearly mean wind speed of the no WF simulations.
The bottom row of Fig. 3 shows the effect of constructing the new Princess Elisabeth wind farm cluster. The wind farm cluster
most affected by the new PE wind farm cluster is the Belgian-Dutch (BE-NL) wind farm cluster (Fig. 1), with differences in
Annual Energy Production (AEP) up to 14% at certain grid cell locations and 5% spatially averaged over the farm (Table 6).
This effect is primarily due to the proximity of this cluster to the Princess Elisabeth wind farms and its downstream location
for the main wind direction, which is southwesterly (Ivanova et al., 2023) . The BE-NL cluster experiences the most significant
impact but also, smaller yet non-negligible effects are observed for wind farms further away. For instance, AEP losses of up to
1% are observed for wind farms shown in Fig. C1. Even wind farms located at a distance of 153 km exhibit a small AEP loss of
0.3% (Fig. C1). Future attention should be given to these low observed differences, as they could potentially stem from random
variation rather than reflecting true effects. It is imperative that further research investigates these discrepancies and assesses
whether such small deviations hold statistical and economic significance. The observed impact on distances underscores the
significance of factoring in AEP losses for more distant wind farms in future siting decisions. This consideration becomes
increasingly crucial as the density of offshore wind farms is projected to increase in the coming years (Borgers et al., 2024).
There should be noted that very small increases (0.1 to 0.2%) in AEP are seen for three wind farms after the build of the PE
wind farm. Due to these low values occurring just at three wind farms it is difficult to draw any strict conclusions on increases
in AEP and further research is recommended.





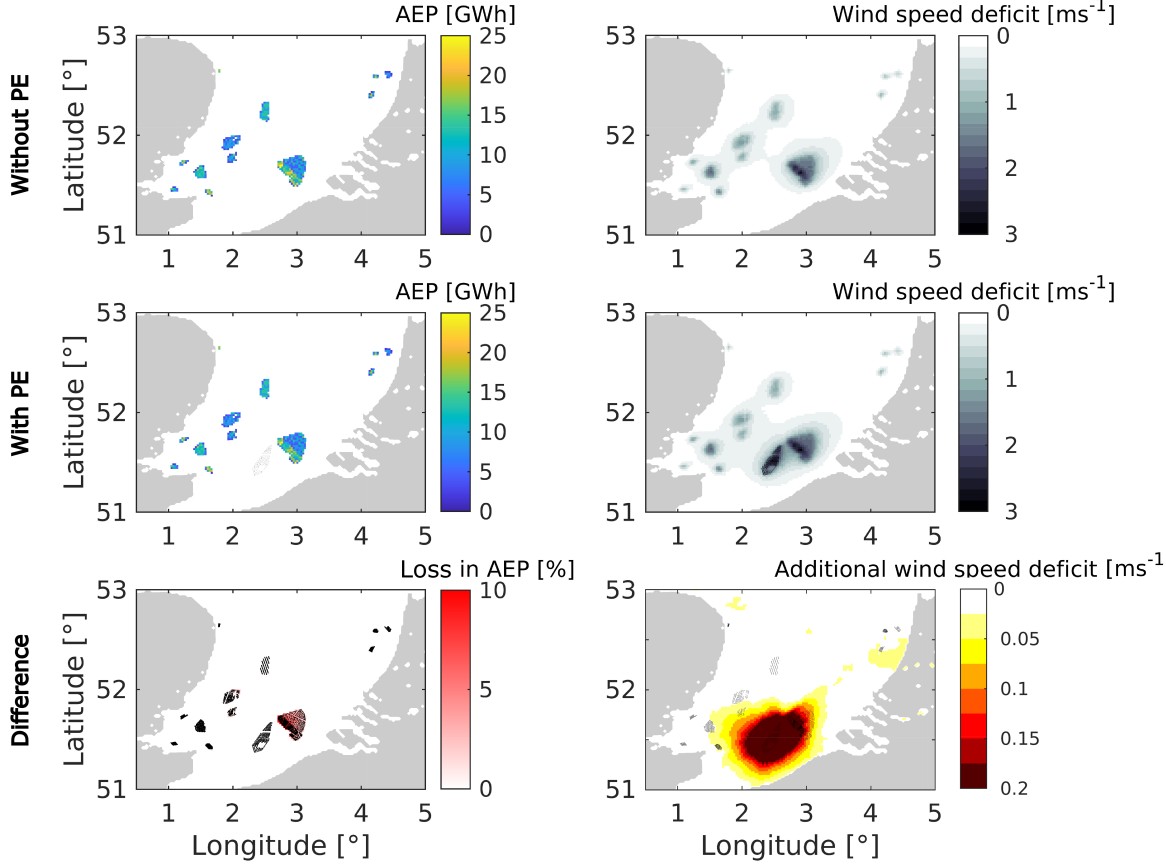

**Figure 3.** Annual energy production (left column), mean yearly wind speed deficit (middle column) fields for the situation before (top row) and after (middle row) the build of the PE wind farm cluster. The last row shows the first row minus the second row.

Due to the presence of the new PE wind farm cluster, wind deficits are intensified (maximum deficit from 2.7 to 2.9 ms$^{-1}$) and spread over a larger region. This effect is attributed to the increased momentum extraction of the 15 MW turbines planned for the PE wind farm cluster. The wind speed deficit exerts the strongest influence in the wake in the main wind direction, which is Southwesterly (Ivanova et al., 2023). However, wind speed deficits are also observed upstream of the new PE farms and extend towards the Northwest of the future concession. These wind speed deficits contribute to the changes in Annual Energy Production (AEP), with the most substantial deficits corresponding to larger AEP losses. It is essential to note that the wind speed deficits mentioned here are yearly averaged deficits, and their strength and direction are strongly dependent on wind direction, stability and less on TKE coefficient (Rosencrans et al., 2024), as we further discuss in the following sections.



**Table 6.** Distance to the PE wind farm cluster (from the center of the PE farm to the center of the considered farm), the external yearly spatial average AEP loss over the specific wind farm, and the external spatial maximum AEP loss within the wind farm cluster.

| | Distance to PE [km] | Yearly and spatially averaged AEP loss [%] | Spatial maximum AEP loss within the farm [%] |
|---|---|---|---|
| BE-NL cluster | 34 | 5.1 | 14.4 |
| Scroby Sands | 135 | -0.1 | -0.1 |
| East Anglia ONE | 81 | 0.1 | 0.5 |
| Galloper & Greater Gabbard | 54 | 0.8 | 3.9 |
| Gunfleet Sands | 89 | -0.1 | 0 |
| London Array | 68 | 0.5 | 0.7 |
| Kentfish Flats | 95 | -0.2 | -0.2 |
| Thanet | 58 | 0.9 | 1.2 |
| Luchterduinen | 153 | 0.3 | 0.4 |
| Egmond aan Zee | 181 | 0.1 | 0.2 |
| Prinses Amalia | 170 | 0.1 | 0.2 |

### 3.3 External wake losses as a function of wind direction, wind speed and atmospheric stability

To gain more insight into the phenomena influencing external wake losses (see Section 2.5, Fig. 4 (a)) a more in-depth study of the BE-NL wind farm cluster (Fig. 1) is conducted. This wind farm zone is investigated because it experiences the largest external wake losses.

     In the following figures, Fig. 4 (b)-(d), Fig. 6 (a), Fig. 7 (b)-(d), and Fig. 10 we characterize the wind conditions (wind direction, atmospheric stability class and wind speed) based on a reference location (spatial averaged grid cell location of the

considered wind farm) of the no WF WRF simulation to quantify the background atmospheric boundary layer conditions in the absence of wind farms. The height at which the wind speed and direction is taken, is at 96 m altitude, which is the weighted averaged hub height over all wind turbines considered in this work. The error bars in these figures represent a 95% confidence interval calculated by bootstrapping with 1000 resamples.

     The external wake losses are concentrated at the top left corner of the BE-NL wind farm cluster, resulting in external wake

losses up to 12.25% at these wind turbine locations (Fig. 4 (a)). These high losses are linked to the prevailing southwesterly wind direction. This is because the new PE wind farm is upstream of the BE-NL cluster for this wind direction (Fig. 4 (b)). This leads to the highest losses at the top left corner of the BE-NL cluster for wind directions of approximately 225° for which the impact of the PE wind farm cluster is the highest (Fig. 5). Even for other wind directions the new PE wind farm cluster causes external wake losses for the BE-NL cluster. Additionally, there is a trend of decreasing external wake losses for very

stable to very unstable regimes, however these result are not statistically significant. The strongest wake losses, even thought not statistically significantly different, are present for very stable atmospheric conditions, while the smallest losses are present



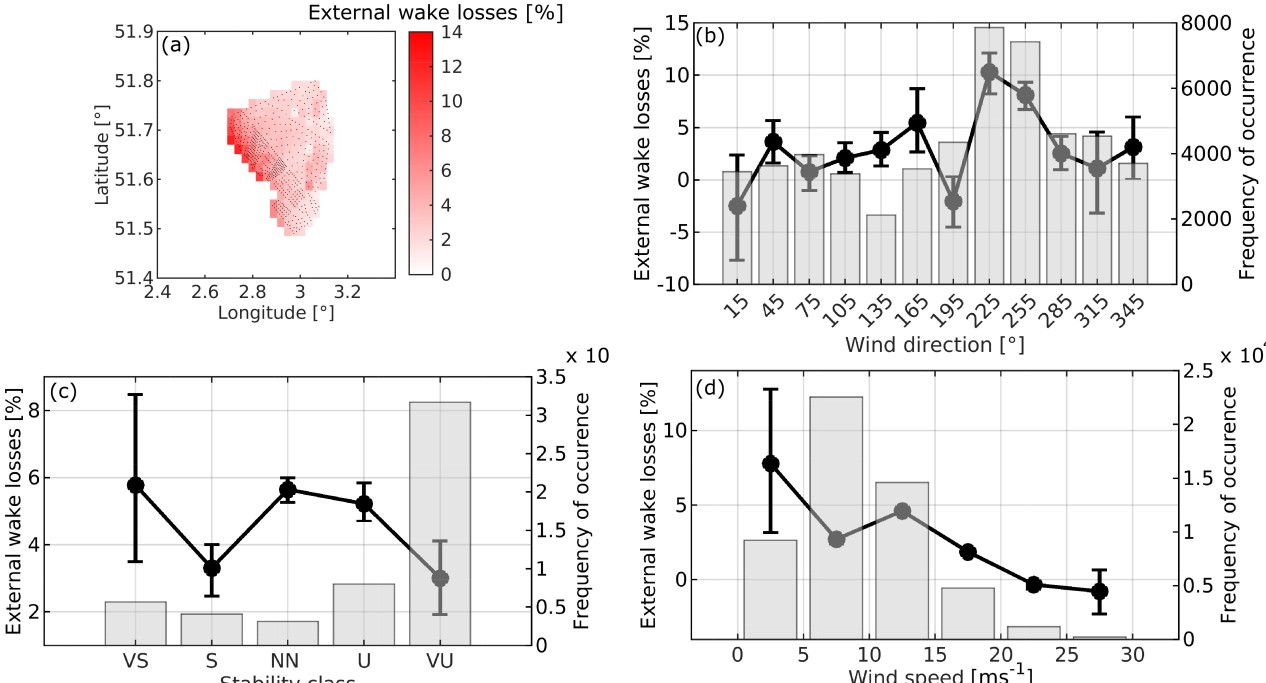

**Figure 4.** Results are shown for the BE-NL wind farm cluster. (a) External wake losses for the BE-NL wind farm. (b) (Line) External wake losses as a function of wind direction, (bars) frequency of occurrence of the different wind directions. (c) (Line) External wake losses as a function of atmospheric stability classes (VS: Very Stable, S: Stable, NN: Near Neutral, U: Unstable, VU: Very Unstable), (bars) frequency of occurrence for the different atmospheric stability classes. (d) (Line) External wake losses as a function of wind speed, (bars) frequency of occurrence of the different wind speeds. The error bars represent a 95% confidence interval. For (b)-(d) the external wake losses are averaged over the whole BE-NL wind farm cluster.

for the most frequently (60.2% of the time) present very unstable atmospheric conditions (Fig. 4 (c)). It has been reported in earlier studies that for the North Sea (very) unstable atmospheric conditions are commonly seen (Porchetta et al., 2019; Munoz-Esparza et al., 2012; Nybø et al., 2020). Also at the location of the planned Vineyard wind farm at the East coast of the
US, unstable conditions are present 53.6% of the time (Rosencrans et al., 2024). Stable conditions in the North Sea are mostly seen for southwest wind direction, when warmer air is transported over the colder sea especially in summer, while other wind directions are dominated by (very) unstable conditions of cold air above the warmer sea especially in winter (Nybø et al., 2020; Cheynet et al., 2018). The external wake losses are dominated by lower wind speeds (Fig. 4 (d)) at which the wind turbine thrust coefficients are higher, resulting in longer recovery times.
Interestingly, negative external wake losses (-3%), i.e. positive interference from the construction of the new PE wind farm cluster, are seen for winds coming from the north or south (Fig. 4 (b)). Even though the confidence interval of these losses is large, a general negative trend is present, resulting in no statistically significant negative wake losses. Looking at a different

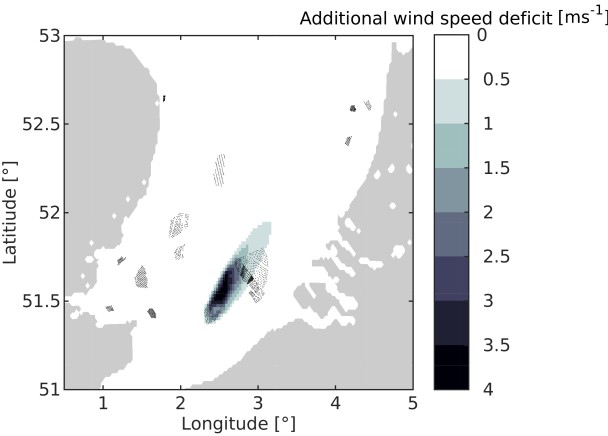

**Figure 5.** External Be-NL wind farm wake deficit for wind directions between 215 and 235°, which corresponds to the highest external wake losses and the most frequent wind direction.

wind farm (Thanet, Fig. 1) with similar negative external wake losses (-3%) but with smaller error bars over larger wind directions compared to the BE-NL wind farm cluster, shows that negative external wake losses are present for winds from the south (165-195°) (Fig. 6 (a)). Some wind directions result in statistically significant negative external wake losses. The external wake losses reach between 2 and 4% for wind directions coming from 75-105°, 25° and 315-345°. This corresponds to the wind direction when the Thanet wind farm is waked by the PE, Kentish Flats and London Array wind farm respectively. The negative external wake losses are not frequently observed in the current simulations, as such these observations are the mean impact of the new PE wind farm cluster. In the present simulations, the negative external wake losses are explained by an intensification of the speedup at the lateral locations of the PE wind farm clusters (Fig. 6 (b)), the most upstream wind farms for this wind direction. It appears that the PE wind farm cluster acts as a windbreak and speeds up the horizontal velocity in the lateral directions. To our knowledge, a similar phenomenon of speedup has been studied for windbreaks, causing vertical speedups (Liu and Stevens, 2021), but not in depth for lateral speedups influencing farm-farm wakes (Hasager et al., 2015; Nygaard and Hansen, 2016). Nygaard and Hansen (2016) mentioned that there are no modeling results showing these kinds of effects on the farm level. However, they did observe this effect based on measured wind turbine data from two offshore wind farms. In contrast, this speedup effect in the lateral direction has been observed for individual wind turbines (Ammara et al., 2022; Araya et al., 2014; Puccioni et al., 2023). In the coming years, it would be beneficial to enhance our comprehension of this acceleration phenomena around wind farms, by idealized simulations for example, however this is outside the scope of this work. Understanding these speedups is important, as they may have a significant impact on external wake losses, necessitating consideration in the planning of high-density offshore regions.



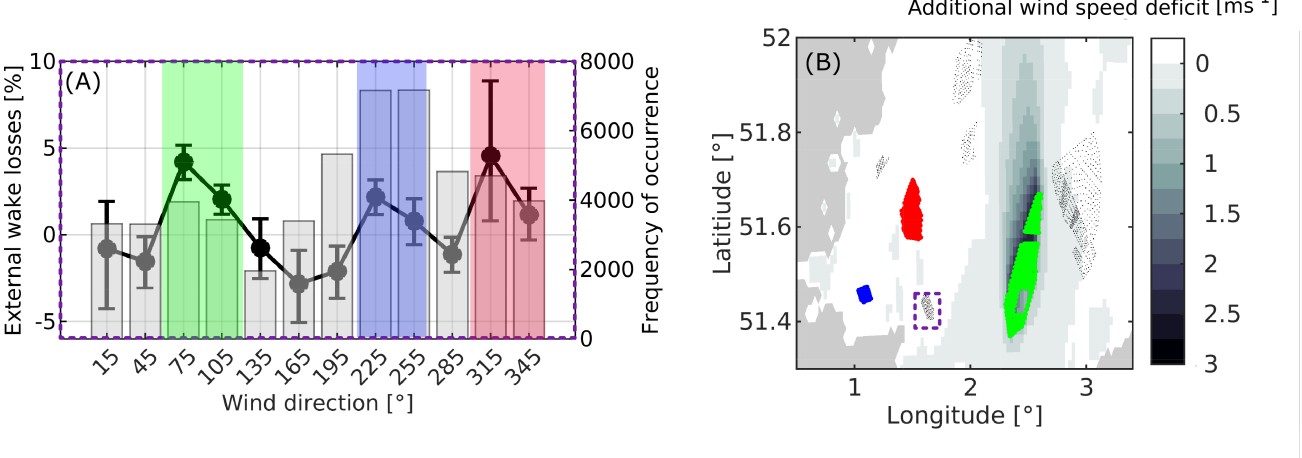

**Figure 6.** Results are shown for the Thanet wind farm (a) (Line) External wake losses averaged over the Thanet wind farm as a function of wind direction, (bars) frequency of occurrence of the different wind directions. The error bars represents a 95% confidence interval. (b) External wake deficit for wind directions between 165 and 195°, which corresponds to the highest negative external wake losses. The colors (green, red and blue) indicate the wind direction when the Thanet wind farm is waked by respectively the PE wind farm cluster, London Array and Kentish Flats.

### 3.4 Internal wake losses as a function of wind direction, wind speed and atmospheric stability

As shown in Table 7, the internal wake losses account for a larger fraction of the total losses. The internal wake losses are greater than 7.5% for all simulated wind farms, and reach as high as 36% for the BE-NL wind farm cluster. These losses are the yearly spatially averaged wake losses. Also the new wind farm deployments at the East coast of the US, even though with a different configuration, are predicted to have similar order of magnitude internal wake losses using the WRF model (Rosencrans et al., 2024; Rybchuk et al., 2022). Additionally, at specific locations within our simulated wind farm the internal wake losses (reaching up to 58% (Fig. 7 (a)) reach similar values as previous studies. These high internal wake losses are associated to very dense wind farm layouts. Unlike the external wake losses (Fig. 4 (b)), which are dominated by wind directions waked by the PE wind farm, the internal wake losses are dominated by the least frequent wind direction (4% of the time) (Fig. 7 (b)). This wind direction corresponds to the wind direction where the largest number of turbines are influenced by upstream turbines. The internal wake losses furthermore decrease from stable to unstable atmospheric conditions and from low to high wind speeds (Fig. 7 (c)-(d)).

We seek to further understand the complex interactions between the internal and external wake effects. To do so, we look at the internal wake losses of five subsequent wind turbines for a specific wind direction in and out of the wake of the new PE wind farm cluster. In Fig. 8, we demonstrate that the structure of the internal wake losses of the selected wind turbines are affected by the new upstream PE wind farm cluster. If the turbines are not in the wake of the upstream PE wind farm cluster the internal wake is unaffected by the presence of the new PE wind farm cluster (Fig. 8 (a)-(e)). However, the internal wakes of the selected wind turbines of the downwind farm that are in the wake of the new PE wind farm cluster undergo non linear





**Table 7.** Yearly and spatially averaged external and internal wake losses over the wind farm(s).

|  | External wake losses [%] | Internal wake losses [%] |
|---|---|---|
| BE-NL cluster | 3.58 | 36.03 |
| Scroby Sands | -0.13 | 19.45 |
| East Anglia ONE | 0.08 | 18.50 |
| Galloper & Greater Gabbard | 1.64 | 19.35 |
| Gunfleet Sands | -0.54 | 19.05 |
| London Array | 0.66 | 28.39 |
| Kentfish Flats | -1.29 | 14.48 |
| Thanet | 0.76 | 25.01 |
| Luchterduinen | 1.08 | 9.29 |
| Egmond aan Zee | 0.25 | 7.54 |
| Prinses Amalia | 0.41 | 18.53 |

effects (Fig. 8 (f)-(j)). The power normalized by the first row turbines is higher for the turbines when they are waked by the

PE wind farm cluster (Fig. 8 (h)). This increase in power could be caused by a higher added TKE compared to the simulation without the new PE wind farm cluster (Fig. 8 (j)) which enhances the mixing of the wake and thus dissipates the wake faster. Additionally, the velocity deficit is smaller for the waked turbines which additionally results in higher power after the front row wind turbine (Fig. 8 (i)). Together, the new PE wind farm cluster lowers the overall power of the turbines (Fig. 8 (g)) but it also reduces the internal wake effects (Fig. 8 (h)). This phenomena again emphasized the caution that should be paid in the

future for farm-farm interactions, as they can induce non-linear effects within a distant wind farm.

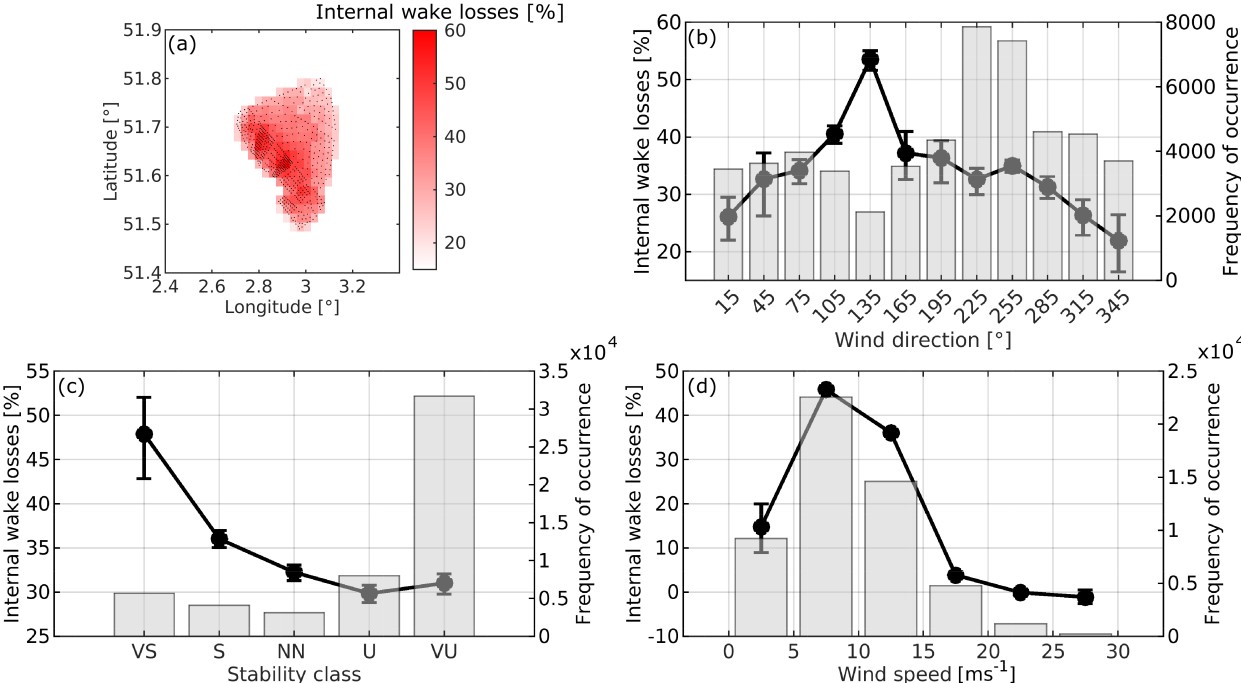

**Figure 7.** Results are shown for the BE-NL wind farm cluster. (a) Internal wake losses for the BE-NL wind farm. (b) (Line) Internal wake losses as a function of wind direction, (bars) frequency of occurrence of the different wind directions. (c) (Line) Internal wake losses as a function of atmospheric stability classes (VS: Very Stable, S: Stable, NN: Near Neutral, U: Unstable, VU: Very Unstable), (bars) frequency of occurrence for the different atmospheric stability classes. (d) (Line) Internal wake losses as a function of wind speed, (bars) frequency of occurrence of the different wind speeds. The error bars represents a 95% confidence interval. For (b)-(d) the internal wake losses are averaged over the whole BE-NL wind farm cluster.

**Figure 8.** (a) & (f) In red are the wind turbine locations considered for a specific wind direction (depicted by the black arrow (a) 105° (f) 234°), for (a) the BE-NL wind farm cluster is not waked by the PE wind farm cluster while for (f) the BE-NL cluster is waked by the PE wind farm cluster. (b) & (g) The wind turbine power per turbine row normalized by the no WF front row turbine for unwaked (blue) and waked (orange) conditions. (c) & (h) The wind turbine power per turbine row normalized by the front row turbine for unwaked (blue) and waked (orange) conditions. (d) & (i) The wind speed deficit (compared to the noWT case) per turbine row normalized by the front row turbine for unwaked (blue) and waked (orange) conditions. (e) & (f) The added TKE (compared to the no WT case) per turbine row normalized by the front row turbine for unwaked (blue) and waked (orange) conditions.



## 3.5 Hourly and monthly variation in external and internal wake losses

Due to the variable atmosphere, the yearly averaged wake losses are not informative on which atmospheric conditions are more critical to induce larger internal or external wake losses. There is a different behaviour, in both hourly and monthly timescales, between the external and internal wake losses (Fig. 9 (b) & (d)). The hourly variation is stronger for external wake losses

compared to internal wake losses, and both vary seasonally. Larger external wake losses are observed in summer during the late afternoon, while the internal wake losses variability are mainly determined by the summer season. The higher wake losses in summer are explained by the lower wind speeds resulting in higher thrust coefficients of the wind turbines (Fig. 9 (e)) and the higher occurrence of stable atmospheric conditions (Fig. 9 (g)), when warm air convects over the colder water (Rosencrans et al., 2024). Opposite, lower wake losses in winter occur for higher wind speed and more unstable atmospheric conditions

(Fig. 9 (e) & (g)), when cold air moves across the warmer water (Rosencrans et al., 2024).



**Figure 9.** Results are shown for the BE-NL wind farm cluster. (a) Yearly averaged external wake losses. (b) Hourly and monthly variations of the spatial averaged external wake losses. (c) Yearly averaged internal wake losses. (d) Hourly and monthly variations of the spatial averaged internal wake losses. Hourly and monthly variations of (e) wind speed, (f) wind direction and (g) occurrence of stable atmospheric conditions.

### 3.6 Comparison of external and internal wake losses between the WRF model and engineering wake models

The two model strategies (WRF-engineering models) to define wake losses in this study differ in their computational costs, methodology and assumptions. As such, a quantitative and a qualitative comparison of their wake loss prediction over a year is made. For the yearly BE-NL wind farm wake losses the models differ more for the external wakes compared to internal wakes (Tab. 8), which can be expected as the engineering models were not designed to predict external wake losses. This due to their much more rapid wind deficit recovery in the far wake (Fischereit et al., 2022b), suggesting that the simplified turbulence






**Table 8.** The external and internal wake losses for the WRF model and the different engineering models together with the percentage difference between WRF and the different engineering models for the BE-NL wind farm cluster.

| | External wake losses [%] | Internal wake losses [%] | Difference to WRF external wake losses [%] | Difference to WRF internal wake losses [%] |
|---|---|---|---|---|
| WRF | 3.58 | 36.03 | | |
| Gauss- BPA | 0.06 | 14.88 | -98 | -59 |
| Jensen k_w=0.02 | 0.64 | 27.87 | -82 | -23 |
| Jensen k_w=0.04 | 0.20 | 19.34 | -94 | -46 |
| Cumulative Curl | 1.70 | 25.54 | -53 | -29 |
| TurbOPark A=0.04 | 2.40 | 30.87 | -33 | -14 |
| TurbOPark A=0.06 | 1.76 | 26.30 | -51 | -27 |

assumption in these models are not sufficient to capture the farm-farm interactions properly. From previous work, for a shorter time span, with validation data, it resulted that the external wake losses are underestimated by the engineering wake models and that they were less suitable to capture the external wakes compared to the predictions of WRF (Fischereit et al., 2022b).

Two of the engineering models (Cumulative curl and TurbOPark) in this work were recently developed to compensate for this underestimation of external wakes (Pedersen et al., 2022; Bay et al., 2023). Indeed, quantitatively the yearly external wake losses of the BE-NL wind farm cluster are closer to the observed wake losses predicted by WRF. The found wake loss difference of 50% of the engineering models compared to WRF by Fischereit et al. (2022b) is reduced to 33% within this work for the TurboPark model ($A$=0.04) (Pedersen et al., 2022). Note that the external wake losses observed in the current study are

significantly larger than the ones reported in Munters et al. (2022). This can be explained by the fact that the latter study is based on the older top-hat implementation of the TurbOPark model, along with the models that produce the lowest losses also in this study (Jensen $k_w$ = 0.04 and Gauss-BPA). It is, moreover also important to notice that the parameter estimates of the engineering models ($k_w$ & $A$) give quantitative differences as large as between different engineering models itself. As currently a truth value is missing in this study, because the PE wind farm cluster still has to be built, it is not straightforward to point out

which model is the most appropriate to estimate yearly power estimates and wake losses. However, it should be stressed that a large spread in wake loss estimates is present between different models and can thus have a substantial impact on decision making of new closely located wind farms.

The variation of the external and internal wake losses with respect to the wind conditions (wind direction, wind speed and atmospheric stability) for the BE-NL wind farm cluster are shown in Fig. 10. For both the external and internal wake losses

the differences between the models are mostly explained by the differences in the wake losses being strongly dependent on the atmospheric stability. This suggest that the different model strategy of turbulence between the models is crucial in estimating the wake losses. The variation of the internal wake losses with wind direction shows less variation for the engineering models compared to the WRF model while for both models the variation with wind speed is well captured. Furthermore, quantitatively the newer engineering models (Cumulative curl and TurbOPark) are closer to the wake loss estimates of WRF, especially the



engineering model of TurbOPark A=0.04 has the closest estimates to the WRF predictions. Moreover, this trend is present for both external and internal wake losses and for all wind conditions.

**Figure 10.** (Left) External and (right) internal wake losses [%] in respect to different wind directions (a) and (b), to different stability classes (c) and (d), and to different wind speeds (e) and (f) for the WRF model and the different engineering models (Gauss-BPA, Jensen ($k_w = 0.02$), Jensen ($k_w = 0.04$), Cumulative curl, TurbOPark ($A = 0.04$), TurbOPark ($A = 0.06$)).

The qualitative yearly internal and external wake losses of the different models are shown in Fig. 11 and 12, respectively. Also the differences between these models are shown. For the internal wake losses the qualitative patterns are similar between the different models. The internal wake losses are concentrated within the farm and are minimal for the wind turbines located at the edge of the BE-NL wind farm cluster. The highest internal wake losses are seen at the location of the most dense wind



farms within the BE-NL wind farm cluster. All models show this results, however all with a different magnitude (Fig. C2). Even though it is known that internal wakes are relatively well estimated by engineering models (Peña et al., 2018), it is possible that due to the size of the large BE-NL wind farm cluster, speed and direction gradients cause some losses that are not well captured by the engineering models, which could explain the large differences compared to the WRF model (Fischereit

et al., 2022b). The qualitative patterns of the yearly external losses (Fig. 12) do not all show the same pattern (Fig. C3). Indeed, even though the magnitude of the external wake losses differs between the models, only the new engineering models (Cumulative curl and TurbOPark) (Pedersen et al., 2022; Bay et al., 2023) do show a more similar pattern to the WRF model, e.i. higher external wake losses in the north west of the wind farm cluster. These engineering models are especially designed to reduce the underestimation of the external wake losses (Bay et al., 2023; Pedersen et al., 2022). There can be seen that

quantitatively the results of these engineering models (Table 8) are also closer to the WRF model. Because a truth value is missing, no conclusion can be drawn about which model performs best, however differences between the models are clear both quantitative and qualitative. There are, however, indications that the WRF model is better equipped in estimating the external wake losses (Fischereit et al., 2022b). Despite this, future work should focus on validating different model strategies for external and internal wake losses over these longer time periods. Furthermore, this work only included the wind farm parameterization

of Fitch et al. (2012), which performs better for external wake losses compared to the wind farm parameterizaion of Volker et al. (2015) (Fischereit et al., 2022b) however the performance of other wind farms parameterizations are not considered so far for the estimate of yearly external wake losses even though it is know that there are quite some variations (Ali et al., 2023). Additionally, the wind farm parameterization of Fitch et al. (2012) has been employed with a TKE coefficient of 0.25, while it is know this influences the wake behaviour of the farms (Rosencrans et al., 2024). Lastly, a horizontal grid resolution of 2 km

has been set in this study, following to Fischereit et al. (2022a), a higher grid resolution could possibly alter the wake estimates.

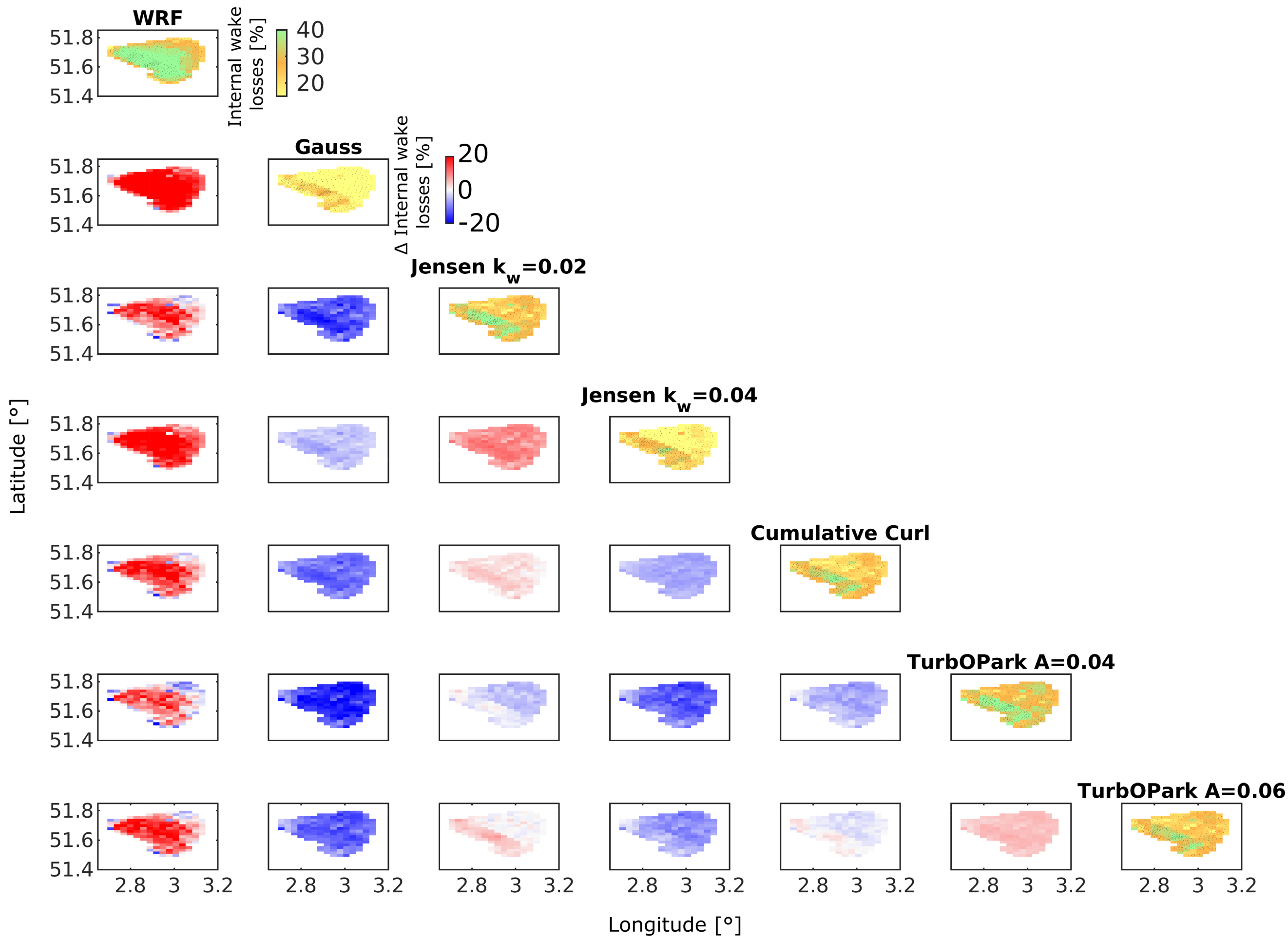

**Figure 11.** On the diagonal the yearly internal wake losses for the WRF model and the different engineering are shown (Gauss-BPA, Jensen ($k_w = 0.02$), Jensen ($k_w = 0.04$), Cumulative curl, TurbOPark ($A = 0.04$), TurbOPark ($A = 0.06$)). Below the diagonal the difference in yearly internal wake loss between the model of that column and row is depicted.



**Figure 12.** On the diagonal the yearly external wake losses for the WRF model and the different engineering are shown (Gauss-BPA, Jensen ($k_w = 0.02$), Jensen ($k_w = 0.04$), Cumulative curl, TurbOPark ($A = 0.04$), TurbOPark ($A = 0.06$)). Below the diagonal the difference in yearly external wake loss between the model of that column and row is depicted.

## 4 Conclusions

The yearly averaged and the sub-annual variation of the wake impacts of a new wind farm cluster on adjacent, existing wind farms is studied, this for the new Princess Elisabeth cluster. The wake impacts of a representative year has been simulated by different model strategies to estimate representative wake losses. These model strategies are a mesoscale model, WRF, and four different engineering wake models (some with different values of the tunable parameters that control wake recovery) for which the wake estimates are compared.

The mesoscale model predicts that building the new Princess Elisabeth wind farm cluster gives differences up to 5% of AEP for a closely located neighbouring wind farm. The external wake losses of this closely located neighbouring wind farm are






mainly determined by the waked wind direction of the Princess Elisabeth wind farm cluster, stable conditions and low wind
speed. The internal wake losses are dominated by wind directions with the highest wind turbine densities within the wind farm.
Additionally, it is clear that the new wind farm cluster influences the internal wakes in a non linear way. Interestingly, some
further located wind farms show a negative wake loss, which is most probably caused by a speedup of the upstream wind farm.
However, a more into depth study of this phenomena is needed.

These internal and external wake losses are investigated by hour and month. The external wake losses exhibit a stronger
hourly variation, while the internal losses depend strongly on season. However, both variations can be linked to the atmospheric
stability and wind speed. With stable atmospheric conditions and lower wind speeds resulting in higher wake losses.

The comparisons of wakes between the different model strategies (numerical weather prediction model vs fast-running
engineering models) results in both quantitative and qualitative differences. The quantitative differences are largest for the
external wake losses compared to the internal losses, from 33% to 98% lower in the engineering models compared to the
mesoscale model, while the internal losses are 14% to 59% lower. Qualitatively the same pattern is present across all models
for the internal wake losses, while this is not the case for the external wake losses. For the internal wake losses, highest losses
are present within the wind farm, with highest losses for the most densely spaced wind turbines, and lower wake losses are
present at the edge of the wind farm. For the external wake losses the qualitative differences are less between WRF and the
newly developed engineering models (Cumulative Curl and TurbOPark) compared to the other engineering models, for which
the former were designed to reduce the underestimation of the external wake losses. It is important to notice that a large spread
in both external and internal wakes is present. This can have a large impact on siting and design decisions. As such, in the near
future more long term validation of the different model strategies is recommended.

This study provides a year-long dataset for both mesoscale models and engineering wake models, which could serve as valu-
able calibration data for future improvements. The wind farm parameterizations within the mesoscale model could be refined
for intra-farm wakes, while the engineering wake model could be improved for farm-farm wakes. Especially as these results
show that the engineering models lack the inclusion of different atmospheric stability classes, and thus a correct turbulence
representation. However, the development of enhanced wake models falls outside the scope of this work. Although FLORIS has
capacities for more advanced setups including effects of, e.g., shear, veer and horizontally heterogeneous background flows,
we deliberately retain a simplified setup in the current study to compare the WRF results to a typical energy yield assessment
practice.

Future work could also include validation data when the new Princess Elisabeth wind farm cluster is build. This would
give a more clear view on the truth values and could accelerate the model developments in both WRF and the fast-running
engineering models.

*Code and data availability.* The Advanced Research WRF (ARW) model is developed by the National Center for Atmospheric Research
(Skamarock et al., 2008). WRF v4.3 is publicly available athttps://github.com/wrf-model/WRF/releases/tag/v4.3. The forcing data used for
WRFs initial and boundary conditions in the WRF simulations is also publicly available at https://cds.climate.copernicus.eu/cdsapp!/dataset/reanalysis-



era5-single-levels?tab=form and https://cds.climate.copernicus.eu/cdsapp!/dataset/reanalysis-era5-pressure-levels?tab=form. Data from the numerical simulations and the namelists used in the WRF model are available upon reasonable requests excluding any potentially confidential information. The FLORIS v3.4 model is available at https://github.com/NREL/floris/releases/tag/v3.4. For the measurement mast data

from he Belgian coast, data are accessible at https://meetnetvlaamsebanken.be/.



## Appendix A: Wind turbine details

**Table A1.** Wind Farm details, including the name of the wind farm, the number of turbines per wind farm, the turbine capacity, the turbine hub height and turbine rotor diameter.

| Wind farm name | Total number of turbines (#) | Turbine Capacity | Hub Height (m) | Rotor Diameter (m) |
|---|---|---|---|---|
| Belwind | 55 | 3.0 MW | 70.1 | 112 |
| Belwind Alstom Hailiade | 1 | 6.0 MW | 98.1 | 150 |
| Borssele I<br>Borssele II | 94 | 8.4 MW | 107 | 164 |
| Borssele III<br>Borssele IV | 77 | 9.5 MW | 107 | 164 |
| Borssele V | 2 | 9.5 MW | 107 | 164 |
| C-POWER I | 6 | 5.0 MW | 93.3 | 126 |
| C-POWER II & III | 48 | 6.15 MW | 93.3 | 126 |
| East Anglia ONE | 102 | 7 MW | 120 | 154 |
| Egmond aan Zee | 36 | 3 MW | 70 | 90 |
| Galloper | 56 | 6 MW | 88 | 154 |
| Greater Gabbard | 140 | 3.6 MW | 78 | 107 |
| Gunfleet Sands | 50 | | | |
| | 48 | 3.6 MW | 78 | 107 |
| | 2 | 6 MW | 84 | 120 |
| Kentish Flats | 45 | | | |
| | 30 | 3 MW | 70 | 90 |
| | 15 | 3.3 MW | 83.6 | 112 |
| London Array | 175 | 3.6 MW | 87 | 120 |
| Luchterduinen | 43 | 3 MW | 81 | 112 |
| Nobelwind | 50 | 3.3 MW | 77.1 | 112 |
| Norther | 44 | 8.4 MW | 107 | 164 |
| Northwind | 72 | 3.0 MW | 80.1 | 112 |
| Northwester 2 | 23 | 9.5 MW | 107 | 164 |
| Prinses Amalia | 60 | 2 MW | 60 | 80 |
| Rentel | 42 | 7.35 MW | 102 | 154 |
| Scroby Sands | 30 | 2.0 MW | 68 | 80 |
| SeaMade (Seastar) | 30 | 8.4 MW | 107 | 164 |
| Seamade (Mermaid) | 28 | 8.4 MW | 107 | 164 |
| Thanet | 100 | 3 MW | 70 | 90 |




## Appendix B: Representative year

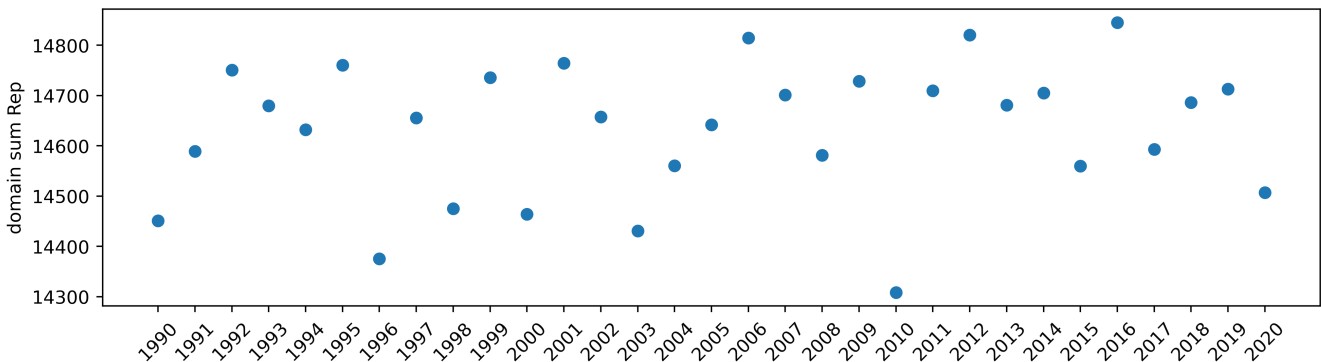

**Figure B1.** Sum of the representativeness per year over the area of interest, where the representativeness is based on the Perkins Skill Score for the wind climate between each single year and the 30 year window (Borgers et al., 2024).

## Appendix C: Additional results

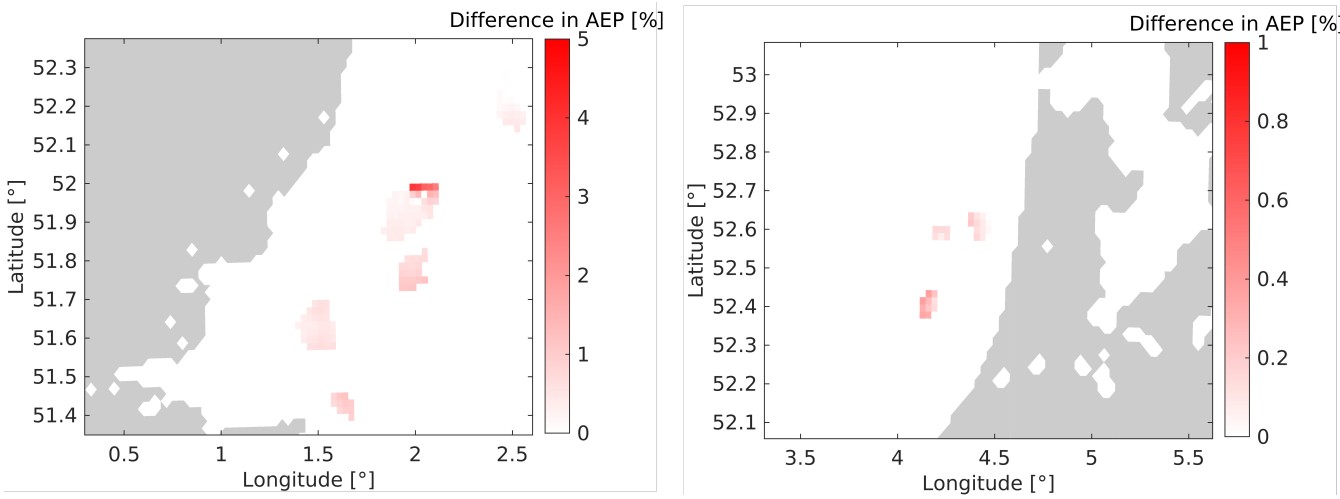

**Figure C1.** The difference in AEP before and after the build of the new Princess Elisabeth wind farm cluster for (left) wind farms in the UK and (right) wind farms in the Netherlands.

**Figure C2.** Yearly internal wake losses of the BE-NL wind farm cluster for the numerical weather prediction model WRF (left) and the fast-running engineering model Gauss-BPA (right).

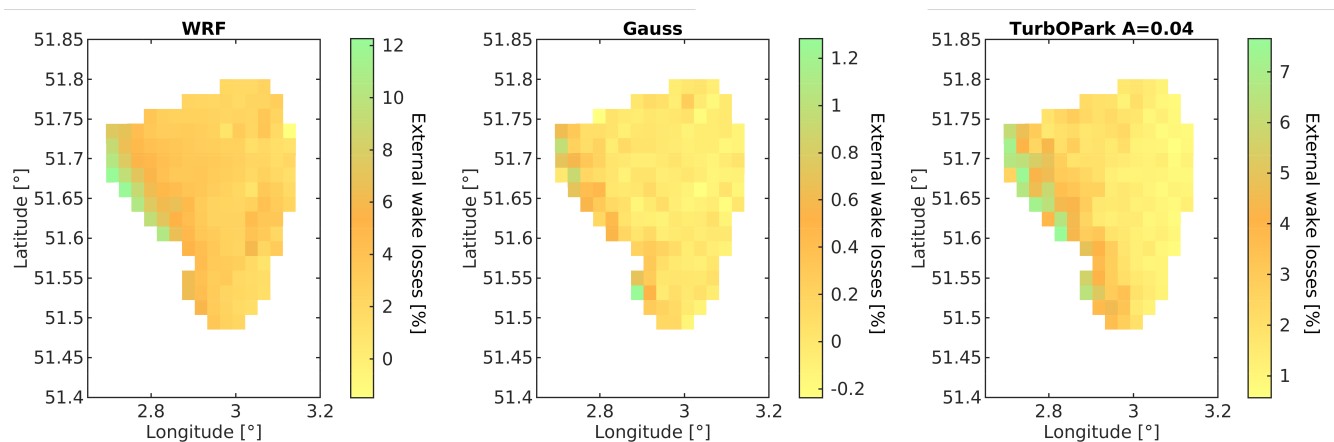

**Figure C3.** Yearly external wake losses of the BE-NL wind farm cluster for the numerical weather prediction model WRF (left) and the fast-running engineering model Gauss-BPA (middle) and the TurbOPark A=0.04 model.

*Author contributions.* SP contributed to the conceptualization, formal analysis, data curation, software, validation, methodology, visualisa-
tion, and writing (original draft, review and editing). MFH contributed to conceptualization, methodology, visualisation, funding, supervision
and writing (review and editing). RB contributed to data curation, methodology, and writing (review and editing). SB contributed to the con-
ceptualization, funding, and writing (review and editing). WM contributed to the conceptualization, formal analysis, validation, methodology,
visualisation, funding, and writing (review and editing).

*Competing interests.* The authors declare that no competing interests are present.





*Acknowledgements.* SP and MFH were funded by MIT Climate Grand Challenges. WM acknowledges support from the Energy Transition Fund of the Federal Public Service Economy of the Belgian Federal Government through the BeFORECAST project, as well as from the Flemish Agency for Innovation and Entrepreneurship (VLAIO) through the Cloud4Wake project. The resources and services used in this work were provided by the VSC (Flemish Supercomputer Center), funded by the Research Foundation-Flanders (FWO) and the Flemish Government.



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
