# Peer review of "Annual wake impacts in and between wind farm clusters modelled by a mesoscale numerical weather prediction model and fast-running engineering models"

_Wind Energy Science, 2024_

## Referee Comment (RC1)

This study aims to provide a quantitative and qualitative comparison of the planned Prinses Elisabeth wind farm cluster across different fidelities of numerical wake models, ranging from mesoscale weather prediction models to fast-running engineering models. However, five of the six sections in the result discussion focus on the mesoscale model, with only the last section discussing the comparison. I believe the authors should make a more detailed analysis comparing the mesoscale model to the engineering model to justify their title. Additionally, the setup and design of the engineering model are not clear to me. Therefore, I don't think the comparison is valid.

**Major Comments:**

**Section 2.4:** More details are needed to describe the setup of the engineering model:

1. How big is the domain? What is the grid resolution? (I assume it is much smaller than the WRF domain, perhaps just the BE-NL wind farm?)
2. "Driving each model with the resulting homogeneous wind fields" – where does the initial wind field come from?
3. "Similar to the WRF case setup, to isolate the effects of the PE wind farm cluster on the existing zone, for every engineering wake model two simulations are performed, i.e., one with and one without the PE." Again, I am curious to know how the wind field is initialized for these simulations.
4. Do you run the engineering model for one year using the wind field from WRF? How do you update the incoming wind field in the engineering model using WRF data?

**Section 3.6:** I believe a more thorough analysis can be done to understand the difference in wake loss between the WRF model and the engineering model. For instance, examining the spatiotemporal differences in wind speed and TKE over the wind farm and its wake region, as well as the wake extent. Differences in the vertical profiles of these variables could provide another perspective. The authors should make a more sincere effort to justify the goal of their paper.

**Minor Comment:**

The definition of internal wake loss (Equation 2) is technically incorrect. While the equation is correct, the definition is flawed due to the design of the experiment. The internal wake loss for a particular wind farm should be the difference between the simulation with and without that particular wind farm only. However, based on how the simulation is set up in this paper, all internal wake losses are contaminated by the external wake loss from other wind farms. Therefore, Table 7 is incorrect.

---

## Author Comment (AC2)

**Review comments**

February 1, 2026

**1 Review 2**

This paper presents a comparison of engineering wake models (available in FLORIS) to the mesoscale WRF simulator for comparing farm-to-farm wakes in the North Sea. Overall, the paper is well-written and its scope is well-defined. The results may help to explain the strengths and weaknesses of different engineering wake models, although some clarity is needed to understand the modeling procedure. I have also left some open questions that the authors may consider addressing (but these should not be treated as necessary). I will add a disclaimer here that I am a developer of the FLORIS engineering wake modeling package that is used throughout this work.

**We would like to thank the reviewer for the suggestions made which significantly improved the manuscript. Following these suggestions and the suggestions of the other reviewers, we decided to add substantial additional analysis comparing WRF and the engineering wake models. However, this would result in a manuscript that would be too long. As such, and at the request of the Editor and the other reviewers, the manuscript has been split into two manuscripts:**

- **Annual wake impacts in and between wind farm clusters - Part I: WRF simulated wake losses for different atmospheric conditions**

- **Annual wake impacts in and between wind farm clusters - Part II: Comparison of WRF and fast-running engineering wake models**

**All reviewer comments will be addressed in the appropriate manuscript (Part I or Part II). Responses to the reviewers' comments are provided in bold, and changes made to the manuscript are highlighted in blue bold italics.**

**Comments on modeling procedure.**

1. The validation procedure for WRF seems fairly limited. As I understood it, there are a handful of locations used for validation, but even the highest point is still less than half of expected turbine hub heights as it is only for a few locations. Moreover, the WRF simulations used for calibration contained no wind farms, since none were built yet; but this does beg the question of how useful such validation is when the rest of the study is comparing wakes. The authors do, to some extent, point out the WRF simulations are not to be treated as "truth"; however, I think the limitations of validation process should be articulated more clearly.

    **We agree that WRF validation was limited and that the limitations of this validation procedure should be articulated more clearly. We have therefore extended the validation by including offshore LiDAR observations providing vertical wind profiles up to hub-height, allowing evaluation of the simulated wind field at heights relevant for wake impacts. Due to the mismatch between the meteorological year and the wind farm operation (PE especially not build yet), a true validation of wake energy losses is not possible and instead, the validation focuses on establishing confidence in the background wind field and atmospheric variability (in between simulations). Accordingly, WRF is not treated as ground truth but as a physically consistent reference for assessing the spread and sensitivity of engineering wake model predictions.**

- **Added LiDAR in Table 2**
- **Added (Part I lines 203-211):**

  *We first seek to establish confidence in the WRF model representation of the wind field. However, we emphasize that the WRF model is not ground truth. The simulations in this study represent idealized scenarios, as only a few of the Belgian-Dutch wind farms (Thorntonbank I, II and III, Belwind, and Northwind, 182 of 572 turbines) were operational in 2016, while the existing turbines used in the simulations correspond to the wind farm layout as of 2022. The year 2016 was nevertheless selected because it was identified as the most meteorologically representative. As a consequence of this mismatch between the meteorological year and the wind farm layout, a true validation of wake energy losses is not possible. As such, validation will only become feasible once post-construction field observations for the fully developed wind farm area become available (Palatos-Plexidas et al., 2025). Future research may then also explore the optimal representation of the wind field by selecting the most appropriate model configuration and parameterizations.*

- **Added (Part I lines 240-252):**

  *The LiDAR at LG was designed specifically for assessing offshore wind resources. It provides vertical wind profiles up to 215 meters, enabling model evaluation at heights comparable to the level of turbine hub heights. Comparing the different WRF simulations with the LiDAR observations (Table 5) reveals a small mean bias across all altitudes and different simulation. The bias for wind speed ranges from 0.04 $ms^{-1}$ at 62 m to $-0.10$ $ms^{-1}$ at 215 m in the no WRF simulation, from -0.08/-0.07 $ms^{-1}$ at 62 m to -0.22/-0.23 $ms^{-1}$ at 215 m in the without PE simulation and with PE simulation, respectivily. Wind direction biases gradually decrease with height, ranging from values around 5° at 62 m to around 1.5° at 215 m. This suggests improved directional agreement at higher altitudes. The mean wind speed and direction bias of different simulations often fall within the observational uncertainty, accounting for model variability and LiDAR measurement error, especially between without PE simulation and with PE simulation. Larger differences are observed when comparing to the no WF simulation, this could be caused by wakes. To conclude, the background wind field can be assumed to be similar across simulations, and justify studying the wake effects in the suggested simulations. Overall, based on all observation comparisons, a good agreement between modeled and observed values at hub height supports using WRF to evaluate wake impacts of the newly built PE on existing wind farms.*

- **Added table (Part I Table 5) here table 1:**

Table 1: Mean wind speed [ms$^{-1}$] and wind direction [°] bias (simulation minus observations) and the 95% confidence interval, which accounts for both model variability and when available measurement uncertainty, for the no WF WRF, without PE, and with PE simulations at different altitudes of the LG LiDAR location (Table 1.

| | Wind speed bias [m s$^{-1}$] | | | Wind direction bias [°] | | |
|---|---|---|---|---|---|---|
| **Altitude [m]** | **no WF** | **without PE** | **with PE** | **no WF** | **without PE** | **with PE** |
| 62 | $0.04 \pm 0.04$ | $-0.08 \pm 0.04$ | $-0.07 \pm 0.04$ | $5.77 \pm 0.32$ | $5.49 \pm 0.33$ | $5.10 \pm 0.32$ |
| 90 | $0.03 \pm 0.04$ | $-0.09 \pm 0.04$ | $-0.08 \pm 0.04$ | $5.06 \pm 0.32$ | $4.81 \pm 0.33$ | $4.39 \pm 0.32$ |
| 115 | $0.01 \pm 0.04$ | $-0.12 \pm 0.04$ | $-0.11 \pm 0.04$ | $4.29 \pm 0.32$ | $4.09 \pm 0.33$ | $3.78 \pm 0.32$ |
| 140 | $-0.01 \pm 0.04$ | $-0.14 \pm 0.04$ | $-0.13 \pm 0.04$ | $3.43 \pm 0.32$ | $3.45 \pm 0.33$ | $3.10 \pm 0.32$ |
| 165 | $-0.04 \pm 0.04$ | $-0.17 \pm 0.04$ | $-0.16 \pm 0.04$ | $2.63 \pm 0.32$ | $2.63 \pm 0.33$ | $2.52 \pm 0.32$ |
| 190 | $-0.07 \pm 0.04$ | $-0.19 \pm 0.04$ | $-0.19 \pm 0.05$ | $1.98 \pm 0.32$ | $1.90 \pm 0.33$ | $1.90 \pm 0.32$ |
| 215 | $-0.10 \pm 0.05$ | $-0.22 \pm 0.05$ | $-0.23 \pm 0.05$ | $1.53 \pm 0.33$ | $1.60 \pm 0.34$ | $1.50 \pm 0.33$ |

2. Similarly, I feel the statement that WRF should not be treated as "truth" should appear more strongly earlier in the paper. Although the authors do state that WRF is not the truth, my

impression is that this message often gets lost in comparisons (in this work and others), which may mean we select parametrizations for engineering models based on faulty criteria.

**We agree with the reviewer's concern. We have revised both Part I and Part II to state more explicitly, and earlier in the papers, that WRF should not be treated as ground truth. Furthermore, Part II now repeatedly highlights that the focus of the analysis is on understanding the spread among modeling approaches, not on selecting parameterizations based on comparisons with WRF.**

- **Added explicitly (Part I lines 203-211):**
  *We first seek to establish confidence in the WRF model representation of the wind field. However, we emphasize that the WRF model is not ground truth.*

- **Added already in the introduction (Part II lines 78-84):**
  *Despite these advances, it remains unclear which modeling approach (numerical weather prediction model vs fast-running engineering wake models) provides more reliable long term estimates of wake impacts for large offshore wind farm clusters. Since no validation data are available within the scope of this study to establish a ground truth, the objective is not to identify a single best-performing model. Instead, comparing wake predictions from different model approaches allows the uncertainty associated with farm–farm wake assessments to be quantified. Such a comparison makes it possible to assess the spread in integrated metrics, such as annual energy production and wake-induced losses, and to identify the atmospheric factors, such as wind speed, wind direction and atmospheric stability that drive differences between model predictions.*

- **Added (Part II lines 267-273):**
  *Based on these findings, TurbOPark with $A = 0.04$ is used as the representative engineering wake model in subsequent analyses when a single wake model is considered. This choice does not imply that TurbOPark provides the most accurate representation of wake effects, given that no observational validation data are available and WRF cannot be regarded as a ground truth.*

- **Added (Part II lines 435-437):**
  *This study investigates how wake losses of offshore wind farm cluster depend on the modelling approach, motivated by the lack of systematic, year-long comparisons between numerical weather prediction models and fast running engineering wake models.*

3. Finally, along this vane, the statement that "advanced" features of FLORIS (heterogeneity, shear(?), veer) appears only in the conclusions (lines 428–430), and I think should have been stated in section 2.4. There, homogeneity is mentioned, but not shear and veer, and since some level of shear is commonly the default configuration for FLORIS (and appears in the example inputs of FLORIS v3.4), this should be stated clearly.

**Thank you for pointing this out, the description of how shear and veer are handled is now present in the methods explaining the engineering wake model setups (section 2.2.2).**

**Added (Part II lines 163-164):**
*No vertical interpolation is applied since both vertical wind shear and veer are omitted in the present study.*

4. The authors ran FLORIS for a grid of wind speeds and wind directions to generate a look-up table of results, which were then mapped to the year-long case. This procedure is reasonable, but I did not quite understand what wind conditions are used to sample the look-up table (line 182). The authors say "the time series extracted from the no WF WRF run," but I assume the wind conditions varied across the domain—how did the authors chose a scalar wind speed and direction to sample the gridded FLORIS results? Were these selected from a single point in the simulation domain, or some sort of average? Lines 381 and 382 suggest (if I'm reading them

correctly) that background flow heterogeneity may be partially the cause of differences, so the choice of representative background flow condition is important here.

The methodology for extracting input to the engineering models from WRF was not entirely clear in the previous manuscript. In the revised version, this methodology is described in greater detail. Furthermore, since the manuscript has been split into two parts, we have also introduced additional setups for the engineering wake models.

Added/Modified (Part II lines 154-202):

*FLORIS simulations are driven by meteorological conditions extracted from the noWF WRF simulations for the entire year 2016 (see Part I). These conditions include wind speed, wind direction, and Turbulence Intensity (TI).*

*The noWF velocity field is provided to FLORIS using two different inflow setups. The first setup assumes horizontally homogeneous inflow and retrieves the wind speed from a single probe point located at the geographic center of the BE−NL cluster at an altitude of 96 m. In contrast, the heterogeneous inflow configuration accounts for spatial variability in the incoming flow, arising from coastal gradients or large-scale synoptic structures.*

*In the heterogeneous configuration, FLORIS' built-in implementation described by Farrel et al. (2021) is used. Specifically, the noWF velocity field at 96 m height is extracted at the WRF grid points and fed to FLORIS. For each wind turbine, FLORIS initializes the local inflow wind speed by performing a linear barycentric interpolation of the WRF wind speed field. Wake velocities downstream of each turbine are then computed following the standard FLORIS wake modeling approach, but using this locally interpolated inflow wind speed as the initial condition. No vertical interpolation is applied since both vertical wind shear and veer are omitted in the present study.*

*Some of the engineering wake model simulations assume a time varying TI estimated based on the noWF WRF data following Larsen (2024). Larsen (2024) proposes to use the Kaimal turbulence spectrum to estimate the wind speed component variances, $\sigma_u$, $\sigma_v$, and $\sigma_w$, at given heights and wind speeds. Empirical ratios between $\sigma_u$ and other components then allow to transform the TKE extracted from WRF into the along-wind standard deviation. The ambient TI can then be approximated as*

$$TI_{amb} = \frac{\sigma_u}{\overline{U}} = \frac{1}{\overline{U}}\sqrt{\frac{2\ TKE}{1 + \alpha^{-1} + \beta^{-1}}} \tag{1}$$

*with $\overline{U}$, the mean wind speed over the averaging period at the height of interest and $\alpha$ and $\beta$, empirical coefficient determined based on Kaimal turbulence spectra.*

*Four different FLORIS simulation setups are subsequently considered:*

- *Homogeneous FLORIS with fixed TI: A time-varying and spatially homogeneous wind field is assumed in FLORIS, constructed from the wind speed and direction at 96 m altitude at the geographic center of the BE−NL wind farm cluster, taken from the no WF WRF simulation (Part I). Turbulence intensity is fixed in space and time at 6%, a typical value for the offshore North Sea area.*

- *Homogeneous FLORIS with varying TI: A time-varying and spatially homogeneous wind field and TI is assumed in FLORIS, constructed from the wind speed, wind direction and TI at 96 m altitude at the geographic center of the BE−NL wind farm cluster, taken from the no WF WRF simulation (Part I). This setup aims at evaluating whether incorporating temporal variability in ambient turbulence resulting from changes in different atmospheric conditions improves the representation of wake recovery.*

- *Heterogeneous FLORIS with fixed TI: A time-varying and spatially heterogeneous wind speed field is assumed in FLORIS, constructed from the wind speed at 96 m altitude taken from the no WF WRF simulation (Part I). Wind direction is kept spatially homogeneous and time-varying, constructed from the no WF WRF simulation at 96 m altitude at the geographic center of the BE–NL wind farm cluster (Part I). Turbulence intensity is fixed in space and time at 6%, a typical value for the offshore North Sea area.*

- *Heterogeneous FLORIS with varying TI: A time-varying and spatially heterogeneous wind speed field is assumed in FLORIS, constructed from the wind speed at 96 m altitude taken from the no WF WRF simulation (Part I). Wind direction is kept spatially homogeneous and time-varying, constructed from the no WF WRF simulation at 96 m altitude at the geographic center of the BE–NL wind farm cluster (Part I). A time-varying and spatially homogeneous TI is assumed, constructed from TI at 96 m altitude at the geographic center of the BE–NL wind farm cluster, taken from the no WF WRF simulation (Part I).*

*All four setups of the engineering wake models are run in a time-series mode with a 30-min resolution for one full year (2016), where each simulation time step is treated as quasi-steady and independent. Consequently, the model does not account for wake advection or temporal effects in wake propagation contrary to WRF. As noted previously, not all wind farms in the North Sea are modeled, for the engineering models used in this study only those belonging to the BE-NL and PE wind farm cluster are included. This reduced domain of not including all North Sea wind farms allows for the simulation of twelve cases (three layouts (no WF, with PE and without PE similar to WRF) combined with four input setups described above) at a reasonable computational cost, while still capturing the dominant inter-farm wake losses of the PE wind farm cluster on the BE-NL wind farm cluster.*

5. Fig. 10. (c) and (d) seems to imply that the FLORIS results depend on the stability class. However, the authors state that turbulence intensity is fixed at 6% throughout for FLORIS simulations. Is there some other dependency on stability that is used, or is the variation in the FLORIS result due to other correlated effects (different wind speed profiles, for example)?

**We agree that this comparison was not sufficiently clear in the original manuscript. In Part II, we now explicitly describe the different setups of the engineering wake models, including the use of turbulence intensity fields obtained from WRF (see also our response to the previous comment). To address this issue directly, we have redone the analysis, and the updated results are presented in Fig. 7 of Part II. These results show that wake models which do not explicitly account for turbulence intensity are invariant with respect to stability class, whereas models such as Cumulative Curl and TurbOPark exhibit a clear dependence on atmospheric stability especially for external wake losses.**

**Added (Part II lines 412-442 and Fig. 7) here Fig. 1:**
*Building on the seasonal analysis from section 3.3.4, an explicit investigation on how the internal and external wake energy losses from the different modeling approaches behave across atmospheric stability classes will be discussed next (Fig. 7). How these energy losses depend on atmospheric stability classes for WRF only have been discussed in section 3.3 and 3.4 in Part I. In general Fig. 7 shows that, the differences between the modeling approached is primarily controlled by the wake modeling formulation rather than by the inflow conditions.*
*For the internal wake energy losses with heterogeneous varying TI inflow conditions (Fig. 7 (a)), the uncertainty ranges associated with the different engineering wake models show partial overlap with the WRF modeling approach, especially during unstable conditions. However, differences between modeling approaches remain during stable conditions. This confirms that there is an increased sensitivity of wake recovery to stability under low turbulence regimes. Under higher turbulence conditions,*

*turbulent mixing accelerates wake recovery, leading to closer agreement between the modeling approaches. The same can be observed from Fig. 7 (b) which isolates the effect of inflow conditions for the TurbOPark model ($A = 0.04$). In both Fig. 7 (a) and Fig. 7 (b), the uncertainty ranges of the engineering wake models and of the different inflow conditions, respectively, largely overlap, making it difficult to identify a single modeling approach or inflow configuration that is consistently closer to WRF for internal wake losses.*

*For the external wake energy losses when comparing the different modeling approaches under heterogeneous varying TI inflow conditions (Fig. 7 (c)), overlap between uncertainty ranges is observed across multiple stability classes and not only under unstable conditions which was the case for internal wake energy losses. However, in contrast to the internal wake energy losses, the engineering wake models themselves no longer overlap consistently over all stability classes, indicating a stronger sensitivity of external wake losses to the underlying wake formulation. In particular, models that explicitly account for turbulence dependent wake recovery and expansion, such as TurbOPark and the Cumulative Curl model, show closer agreement with WRF across stability classes.*

*When isolating the effect of inflow conditions for the TurbOPark model (Fig. 7(d)), the uncertainty ranges associated with the different inflow configurations overlap with each other, similar as for the internal wake energy losses (Fig. 7(b)). Nevertheless, a slightly closer agreement with WRF is found for varying TI inflow conditions compared to fixed TI inflow conditions for stable atmospheric conditions.*

*Overall, this analysis provides a physical explanation for the seasonal patterns discussed in Section 3.3.3, where larger differences between modeling approaches were observed during summer. The increased occurrence of stable atmospheric conditions in summer enhances the sensitivity of wake recovery to the specific wake formulation, particularly for external wakes. As a result, discrepancies between WRF and engineering wake models, and among the engineering models themselves, become more pronounced under these conditions, whereas during periods dominated by unstable stratification the enhanced turbulent mixing leads to closer agreement across modeling approaches.*

[Figure]

*Figure 1: Spatiotemporal conditional averages of wake energy losses by atmospheric stability class (VS: very stable, S: stable, NN: near-neutral, U: unstable, VU: very unstable). (a,b) Internal wake energy losses for wind speeds between 10 and 15 ms$^{-1}$. (c,d) External wake losses for wind directions from 193 - 317. Lines show engineering wake models for heterogeneous inflow and varying TI and WRF; (a,c) all models, (b,d) TurbOPark (A = 0.04) for all inflow setups and WRF. Bars indicate the frequency of occurrence of each stability class. The error bars represent a 95% confidence interval obtained via bootstrapping with 1000 resamples.*

**Open questions (again, these do not need to be directly addressed in the article, but came to mind as I was reading the text and could perhaps be used to clarify conclusions)**

1. To what extent do the authors feel that retuning certain parameters (e.g. the A of the TurboPark model) could have corrected differences between WRF and FLORIS, not only for the single AEP prediction but also for the time-step to time-step comparison? Is it reasonable to assume that engineering wake models will need to be tuned for specific scenarios? This is touched on briefly in lines 93–96 but not really returned to.

**Re-calibrating parameters (e.g., the wake recovery parameter in TurboPark) could potentially reduce the mismatch between WRF and the engineering wake models, both for integrated metrics such as AEP and for time step to time step comparisons. Wake recovery is strongly influenced by site characteristics (e.g., coastal effects and upstream conditions) and atmospheric stability, and regime-dependent tuning (e.g., by wind direction or stability class) could therefore improve agreement. In engineering wake models, a single expansion or recovery parameter aggregates multiple physical processes, making it unlikely that one parameter setting can consistently**

capture all conditions. Moreover, calibration is inherently site-specific, which limits its applicability for wind farms that do not exist (e.g., PE) where no operational data are available.

Our goal in comparing WRF and engineering wake models is to elucidate the differences for 'out-of-the-box' and widely-used setups in the models. Future work should investigate the influence of parameter uncertainty, both in WRF and engineering models.

2. The mechanism for further downstream farms might be experiencing "negative" wake losses wasn't clearly explained to me. I understand that the authors are not attempting to investigate this phenomenon closely; however, I didn't really understand the candidate explanation in lines 327–329. I have thought about this (even ignoring higher turbulence) as being explained by a change (decrease) in thrust at the front row turbines (due to the presence of an upstream farm causing lower velocities at the front row turbines), which causes a shallower wake and therefore (in certain cases) more power at the downstream turbines. Is that equivalent to what you are saying?

In the revised manuscript we calculated the energy losses based on power averaged and not on efficiency averages. Additionally we added uncertainty intervals for the AEP losses and the energy losses. For the AEP the negative losses can probably be attributed to numerical uncertainties, for the external wake losses the negative values fluctuate around 0 and thus is not a significant result. The differences between the two methodologies AEP and wake losses is also explained (see below). The negative wake losses for the BE-NL wind farm cluster for some wind directions is probably linked to lateral speedups. In Part II of the paper the negative losses are linked to the heterogeneous setups for engineering wake models not including explicitly variations in TI. But as mentioned by the reviewer, this would need further research that is not in the scope of these manuscripts.

- **Clarified (Part I lines 273-278):**
  *It should also be noted that very small increases (0.11-0.15%) in AEP are seen for three wind farms after the construction of the PE wind farm. Given the low magnitude and the fact that these occur at only three locations, it is difficult to draw any strict conclusions regarding actual increases in AEP. These increases in AEP may be attributable to numerical variability, possibly related to coastal effects or model variability in WRF, and further research would be needed to determine whether they have physical (flow speedup) or economic significance.*

- **Added (Part I lines 312-319):**
  *Comparing the external wake energy losses (Table 7) with the AEP losses (Table 6) shows that both tables exhibit similar spatial trends. The BE–NL wind farm cluster, Galloper & Greater Gabbard, London Array, Thanet, and Luchterduinen show the largest impacts, while more distant farms show much smaller effects. Differences between the two tables arise because the metrics are defined using different normalizations. The AEP losses are computed as relative changes between the without PE and with PE simulations and normalized by AEP without PE, whereas the external wake losses are defined as ratios of temporally averaged power normalized by the noWF simulation (Eqs. 4–6). This difference in normalization, together with the implicit weighting of high-production periods in the AEP calculation, explains why some farms show statistically significant AEP impacts while their external wake losses are not statistically significant.*

- **Clarified (Part I lines 363-379):**
  *Interestingly, negative external wake energy losses (-1%), i.e., positive interference from the construction of the new PE wind farm cluster, are observed for winds coming from the north to northeast (0-60) or south (180-210) (Fig. 4 (b)). The negative external wake energy losses for wind directions from the south can be explained by a speedup at the lateral locations of the PE wind*

*farm clusters (Fig. 8). It appears that the PE wind farm cluster act as a windbreak, causing global blockage and accelerating the flow around sides of the wind farm. A comparable lateral acceleration caused by global blockage is reported for certain North Sea wind farm clusters (Borgers et al., 2025). While similar effects have been studied for windbreaks, where they mainly cause vertical speed-ups (Liu and Stevens, 2021), lateral flow accelerations influencing farm-to-farm wakes have only recently been examined in more detail (Hasagar et al., 2015; Nygaard and Hansen, 2016; Borgers et al., 2025). Nygaard and Hansen (2016) observed this farm effect based on measured wind turbine data from two offshore wind farms. This speedup effect in the lateral direction has been observed multiple times for individual wind turbines (Ammara et al., 2022; Araya et al., 2014; Puccioni et al., 2023). In the coming years, it would be beneficial to enhance our understanding of this acceleration phenomenon around wind farms, through idealized simulations, however, this is outside the scope of this work. Understanding these speedups is important, as they may have a significant impact on external wake energy losses, necessitating consideration in the planning of high-density offshore regions. Why negative external wake energy losses are observed for wind directions from the north to northeast is less clear. As mentioned previously for the AEP losses, these may result from numerical variability or other uncertainties in the simulation. The underlying mechanisms of these minor effects, which might be physical or not, should be investigated in the future.*

- **Added figure (Part I Fig 8), here Fig 2.:**

[Figure]

Figure 2: External BE-NL wind farm wake deficit for wind directions between 180 and 210, the sector showing negative external wake losses. This map shows the additional wind speed deficit after the construction of the PE wind farm for timesteps with negative external wake energy losses over the farm.